# Long-range Meta-path Search on Large-scale Heterogeneous Graphs

**Chao Li[1,2,∗], Zijie Guo[1,3,∗], Qiuting He[1], Kun He[1,†]**
[1]School of Computer Science and Technology, Huazhong University of Science and Technology
[2] China Mobile Information Technology Co.,Ltd.
[3] School of Computer Science, Fudan University
`{chaoli,zijieguo,brooklet60}@hust.edu.cn`

## Abstract

Utilizing long-range dependency, a concept extensively studied in homogeneous graphs, remains underexplored in heterogeneous graphs, especially on large ones, posing two significant challenges: Reducing computational costs while maximizing effective information utilization in the presence of heterogeneity, and overcoming the over-smoothing issue in graph neural networks. To address this gap, we investigate the importance of different meta-paths and introduce an automatic framework for utilizing long-range dependency on heterogeneous graphs, denoted as *Long-range Meta-path Search through Progressive Sampling* (LMSPS). Specifically, we develop a search space with all meta-paths related to the target node type. By employing a progressive sampling algorithm, LMSPS dynamically shrinks the search space with hop-independent time complexity. Through a sampling evaluation strategy, LMSPS conducts a specialized and effective meta-path selection, leading to retraining with only effective meta-paths, thus mitigating costs and over-smoothing. Extensive experiments across diverse heterogeneous datasets validate LMSPS's capability in discovering effective long-range meta-paths, surpassing state-of-the-art methods. Our code is available at `https://github.com/JHL-HUST/LMSPS`.

## 1   Introduction

Heterogeneous graphs are widely used for modeling multiple types of entities and relationships in complex systems by various types of nodes and edges. For instance, the large-scale academic network, `OGBN-MAG`, contains multiple node types, *i.e.*, Paper (P), Author (A), Institution (I), and Field (F), as well as multiple edge types, such as Author $\xrightarrow{\text{writes}}$ Paper, Paper$\xrightarrow{\text{cites}}$Paper, Author $\xrightarrow{\text{is affiliated with}}$ Institution, and Paper$\xrightarrow{\text{has a topic of}}$Field. These elements can be combined to build higher-level semantic relations called meta-paths [44]. For example, APA is a 2-hop meta-path representing the co-author relationship, and PFPAPFP is a 6-hop meta-path related to long-range dependency.

Utilizing long-range dependency is essential for graph representation learning. For homogeneous graphs, many graph neural networks (GNNs) [28, 4, 1, 54] have been developed to gain benefit from long-range dependency. Utilizing long-range dependency is also crucial for heterogeneous graphs. For instance, the Internet movie database (`IMDB`) contains $21K$ nodes with only $87K$ edges. Such sparsity means each node has only a few directly connected neighbors and requires models to enhance the node embedding from long-range neighbors. The main challenge in using long-range dependency on heterogeneous graphs is how to alleviate costs while striving to effectively utilize information in exponentially increased receptive fields, which is much more challenging compared to homogeneous

---

[∗]The first two authors contribute equally.
[†]Corresponding author.

38th Conference on Neural Information Processing Systems (NeurIPS 2024).

graphs due to the heterogeneity. In addition, the well-known over-smoothing issue [30, 24] occurring in many GNNs also needs to be addressed.

Heterogeneous Graph Neural Networks (HGNNs) are popular deep learning techniques for heterogeneous graph representation learning. The key idea is to aggregate valuable neighbor information based on a range of relations to enhance the semantics of vertices. Traditional HGNNs are typically classified into two categories: metapath-free methods and metapath-based methods. Most metapath-free HGNNs [63, 18, 21, 34] utilize information from $l$-hop neighborhoods by stacking $l$ layers. However, on large-scale heterogeneous graphs, the number of nodes in receptive fields grows exponentially with the number of layers, making them hard to expand to large hops. A recent work [36] employs the graph transformer [54] to learn heterogeneous graphs, which can only exploit long-range dependency in small datasets due to the quadratic complexity of transformer [17].

Metapath-based HGNNs [48, 11, 23, 52] obtain information from $l$-hop neighborhoods by utilizing single-layer structures and meta-paths with maximum hop $l$, *i.e.*, all meta-paths no more than $l$ hops. Benefiting from the feature of selective aggregation of neighbors based on meta-path, metapath-based HGNNs show greater potential in handling large-scale datasets [55, 52]. For metapath-based HGNNs, to exploit long-range dependency, the maximum hop needs to be large enough as there are no stacking layers. However, the number of meta-paths also grows exponentially with maximum hop $l$, corresponding to exponential receptive fields associated with layers in metapath-free methods. For instance, on `OGBN-MAG`, to gain a 3-hop meta-path based on the 2-hop meta-path PAP, the next node type has three choices (A, P, and F). For each additional hop, the number of possible meta-paths increases exponentially. Hence, utilizing long-range dependencies on large-scale heterogeneous graphs has not been resolved yet.

This paper investigates the importance of various meta-paths and makes two key observations: (1) *A small number of meta-paths dominate the performance*, and (2) *certain meta-paths can have a negative impact on performance*. The second observation explains why few HGNNs gain benefit from long-range neighbors. Different from homogeneous graphs, messages on heterogeneous graphs can be noisy or redundant for specific tasks, and the presence of long-range dependencies makes it more challenging to exclude negative heterogeneous information, even with the use of attention mechanisms. For example, focusing too much on Institution instead of related Papers is harmful to predicting a paper's field. These observations highlight the opportunity to leverage long-range dependencies by selectively utilizing effective meta-paths.

Motivated by the observations mentioned above, we propose a novel method called *Long-range Meta-path Search through Progressive Sampling* (LMSPS). LMSPS focuses on meta-path search and aims to reduce the exponentially growing meta-paths to a subset that is specifically effective for the given dataset and task. LMSPS builds a comprehensive search space initially, including all meta-paths related to the target nodes. Then, it adopts a progressive sampling algorithm to reduce the search space. Finally, LMSPS selects the top-$M$ meta-paths from the reduced search space based on a sampling evaluation strategy. This search stage reduces the exponential number of meta-paths to a constant for retraining. Experimental results on both real-world and manual sparse large-scale datasets demonstrate that LMSPS outperforms existing state-of-the-art methods for heterogeneous graph representation learning.

Our main contributions are summarized as follows:

- We propose a novel meta-path search framework termed LMSPS, which to our knowledge is the first HGNNs to utilize long-range dependency in large-scale heterogeneous graphs.
- To search for effective meta-paths efficiently, we introduce a novel progressive sampling algorithm to reduce the search space dynamically and a sampling evaluation strategy for meta-path selection.
- Moreover, the searched meta-paths of LMSPS can be generalized to other HGNNs to boost their performance.

## 2   Preliminaries

**Heterogeneous graph** [43]. *A heterogeneous graph is defined as $\mathcal{G} = \{\mathcal{V}, \mathcal{E}, \mathcal{T}, \mathcal{R}, f_{\mathcal{T}}, f_{\mathcal{R}}\}$ with $|\mathcal{T}+\mathcal{R}|>2$, where $\mathcal{V}$ denotes the set of nodes, $\mathcal{E}$ denotes the set of edges, $\mathcal{T}$ is the node-type set and $\mathcal{R}$ the edge-type set. Each node $v_i \in \mathcal{V}$ is maped to a node type $f_{\mathcal{T}}(v_i) \in \mathcal{T}$ by mapping function $f_{\mathcal{T}} : \mathcal{V} \rightarrow \mathcal{T}$. Similarly, each edge $e_{t \leftarrow s} \in \mathcal{E}$ ($e_{ts}$ for short) is mapped to an edge type $f_{\mathcal{R}}(e_{ts}) \in \mathcal{R}$ by mapping function $f_{\mathcal{R}} : \mathcal{E} \rightarrow \mathcal{R}$.*

**Meta-path** [44]. *A meta-path $P$ is a composite relation that consists of multiple edge types,* i.e., $P \triangleq c_1 \xleftarrow{r_{12}} c_2 \cdots c_{l-1} \xleftarrow{r_{(l-1)l}} c_l$ *($P = c_1 \cdots c_l$ for short), where $c_1, \ldots, c_l \in \mathcal{T}$ and $r_{12}, \ldots, r_{(l-1)l} \in \mathcal{R}$.*

A meta-path $P$ corresponds to multiple meta-path instances in the underlying heterogeneous graph. For example, meta-path APA corresponds to all paths of co-author relationships on the heterogeneous graph. Using meta-paths means selectively aggregating neighbors on the meta-path instances.

## 3   Related Works

**Heterogeneous Graph Neural Networks.** HGNNs are proposed to learn rich and diverse semantic information on heterogeneous graphs. Several HGNNs [29, 23, 34, 36] have involved high-order semantic aggregation. However, their methods are not applied to large-scale datasets due to high costs. Additionally, many HGNNs [56, 21, 48, 23] have implicitly learned meta-paths by attention. However, few work employs the discovered meta-paths to produce final results, let alone generalize them to other HGNNs to demonstrate their effectiveness. For example, GTN [56] and HGT [21] only list the discovered meta-paths. HAN [48], HPN [23] and MEGNN [5] validate the importance of discovered meta-paths by experiments not directly associated with the learning task. GraphMSE [31] is the only work that shows the performance of the discovered meta-paths. However, they are not as effective as the full meta-path set. So, their learned meta-paths are not effective enough.

**Meta-structure Search on Heterogeneous Graphs.** Recently, some works have attempted to utilize neural architecture search (NAS) to discover meta-structures. GEMS [15] is the first NAS method on heterogeneous graphs, which utilizes an evolutionary algorithm to search meta-graphs for recommendation tasks. DiffMG [8] searches for meta-graphs by a differentiable algorithm to conduct an efficient search. PMMM [27] performs a stable search to find meaningful meta-multigraphs. However, meta-path-based HGNNs are mainstream methods [41, 57, 48, 11], while meta-graph-based HGNNs are specialized. So, their searched meta-graphs are extremely difficult to generalize to other HGNNs. RL-HGNN [60] proposes a reinforcement learning (RL)-based method to find meta-paths. On recommendation tasks, RMS-HRec [37] also proposes an RL-based meta-path selection strategy to discover meta-paths. Both of them are very time-consuming.

## 4   Motivation of Long-range Meta-path Search

Long-range meta-paths can complete the missing information that can not be obtained from close nodes. Take the meta-path MDMDMK (M←D←M←D←M←K) from IMDB as an example. IMDB includes four different entity types: Movies (M), Directors (D), Keywords (K), and Actors (A). The task is to predict the category of the target movies. MDMDMK is a 5-hop meta-path that is hard for experts to understand and then apply. However, for many movies without keywords, the meta-path M←D←M←D←M←K is important because the target movies can aggregate the keyword information from the movies of co-directors.

In addition, SeHGNN [52] employs attention mechanisms to fuse all the target-node-related meta-paths and outperforms the existing HGNNs. SeHGNN has an important observation that models with single-layer structures and long meta-paths outperform those with multi-layers and short meta-paths, indicating the advantages of long-range meta-paths. However, because the number of meta-paths increases exponentially with maximum hops, SeHGNN has to use a small maximum hop to save memory and reduce costs. For example, The maximum hop is 2 for large-scale datasets `OGBN-MAG` [20], which is insufficient, as shown in Table 2 of the experiments. This inspires us to consider whether we can only use effective meta-paths instead of all meta-paths to reduce the consumption of large maximum hops.

We analyze the importance of different meta-paths for SeHGNN on two widely-used real-world datasets `DBLP` and `ACM` from HGB [34]. All results are the average of 10 times running with different random initialization. As the number of meta-paths exponentially increases with the maximum hop, in exploratory experiments, we set the maximum hop $l = 2$ for ease of illustration. Then the meta-path sets are {A, AP, APA, APT, APV} on `DBLP`, and {P, PA, PC, PP, PAP, PCP, PPA, PPC, PPP} on `ACM`.

In each experiment on `DBLP`, we remove one meta-path and compare the performance with the result of leveraging the full meta-path set to analyze the importance of the removed meta-path. As shown

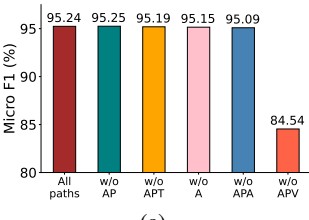 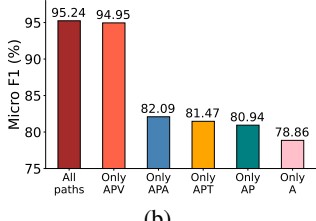 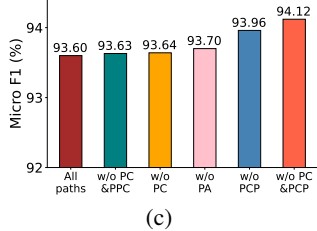

|(a)|(b)|(c)|

Figure 1: Analysis of the importance of different meta-paths. (a) illustrates the results after removing a single meta-path on DBLP; (b) shows the performance of utilizing a single meta-path on DBLP; (c) illustrates the performance after removing a part of meta-paths on ACM.

in Figure 1 (a), removing either A or AP or APA or APT has little impact on the final performance. However, removing APV results in severe degradation in performance, demonstrating APV is the critical meta-path on DBLP when $l = 2$. We further retain one meta-path and remove others as shown in Figure 1 (b). The performance of utilizing APV is only slightly degraded compared to the full meta-paths set. Consequently, we obtain the first observation: *A small number of meta-paths provide major contributions.* In each experiment on ACM, we remove some meta-paths to analyze their impact on the final performance. Results in Figure 1 (c) show that the performance of SeHGNN improves after removing a part of meta-paths. For example, after removing PC and PCP, the Micro-F1 scores are improved by $0.52\%$. So, we can conclude the second observation: *Certain meta-paths can have a negative impact for heterogeneous graphs.* The second observation is reasonable because heterogeneous information is not consistently beneficial for various tasks compared to homogeneous information. It is supported by the fact that various recent HGNNs [8, 27, 52] have removed some edge types to exclude corresponding heterogeneous information during pre-processing based on substantial domain expertise or empirical observations. This observation explains why most HGNNs use a maximum hop of 2, *i.e.*, it is hard to exclude negative information under larger maximum hops. Additionally, because SeHGNN employs an attention mechanism, the performance degradation indicates the attention mechanism has limitations in dealing with noise.

Although long-range meta-paths outperform short meta-paths [52], they need a large maximum hop, resulting in exponentially increasing meta-paths. Motivated by the above two observations, unlike existing methods [5, 31, 52, 56], we can employ effective meta-paths instead of the full meta-path set without sacrificing performance. Although the number of meta-paths grows exponentially with the maximum hop, the proportion of effective meta-paths is small, which is similar to the $80/20$ rule of Pareto principle [40, 10]. To keep efficiency while trading off the performance, we choose a fixed number of meta-paths (like 30) over all datasets. Therefore, the exponential meta-paths can be reduced to a constant. Now the key point is how to find effective meta-paths.

## 5 The Proposed Method

The key of our LMSPS is to utilize a search stage to reduce the exponentially increased meta-paths to a subset effective to the current dataset and task. It can overcome the main challenges of utilizing long-range dependency on heterogeneous graphs. First, utilizing effective meta-paths instead of all meta-paths alleviates computational and memory costs while keeping effective heterogeneous information. Second, each target node only aggregates neighbors on the path instances of effective meta-paths. Because the proportion of effective meta-paths is small and each node has a different neighborhood condition, each target node aggregates different neighbors under the constraints of effective meta-path instances. In this way, the over-smoothing issue can also be overcome.

Then, the main challenge becomes how to discover effective meta-paths, especially long-range ones. Searching for long-range meta-paths has two main challenges: the exponentially increasing issue and the noise issue. We propose a progressive sampling algorithm and a sampling evaluation strategy to respectively overcome the two challenges. Figure 2 illustrates the overall framework of LMSPS, which consists of a super-net in the search stage and a target-net in the training stage.

The super-net aims to automatically discover effective meta-paths for specific datasets or tasks, so the search results should not be affected by specific modules. Based on this consideration, we develop a simple MLP-based instead of transformer-based architecture for meta-path search because the

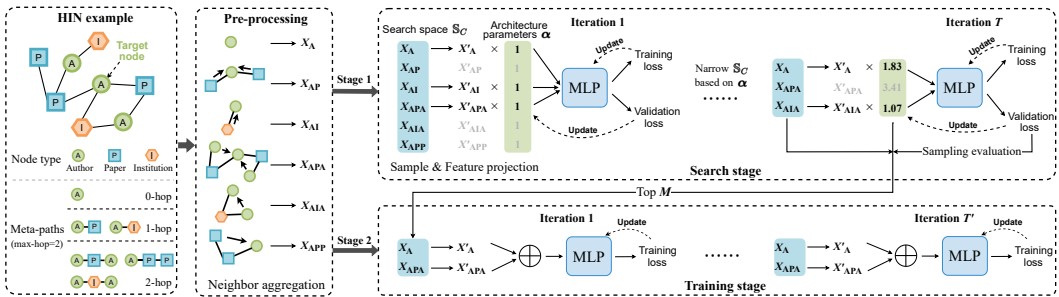

Figure 2: The overall framework of LMSPS. Based on the progressive sampling and sampling evaluation in the search stage, the training stage employs $M$ effective meta-paths instead of the full $K$ target-node-related meta-paths. It exhibits aggregation of meta-paths with maximum hop 2, *i.e.*, 0, 1, and 2-hop meta-paths. The weight updates of feature projection are not shown for ease of illustration.

former involves fewer human interventions [45, 16, 3]. In addition, the sampling strategy keeps the parametric modules changing in the search stage, which is important for preventing the search meta-paths from being affected by specific modules. The super-net contains five blocks: neighbor aggregation, feature projection, progressive sampling search, sampling evaluation, and MLP.

## 5.1 Progressive Sampling Search

Let $\mathbb{P} = \{P_1, \cdots, P_k, \cdots, P_K\}$ be the initial search space with all the $K$ target-node-related meta-paths, $\boldsymbol{X}^{c_i}$ be the raw feature matrix of all nodes belonging to type $c_i$, and $\hat{\boldsymbol{A}}_{c_i,c_{i+1}}$ be the row-normalized adjacency matrix between node type $c_i$ and $c_{i+1}$. The neighbor aggregation block follows SeHGNN [52], which employs an efficient one-time message passing to pre-process an entire heterogeneous graph into regular-shaped tensors for target nodes. Specifically, it uses the multiplication of adjacency matrices to calculate the final contribution weight of each metapath-based neighbor to targets. As shown in Figure 2, the neighbor aggregation process of $l$-hop meta-path $P_k = c_0 c_1 c_2 \ldots c_l$ is:

$$\boldsymbol{X}_k = \begin{cases} \boldsymbol{X}^{c_0} & l = 0 \\ \hat{\boldsymbol{A}}_{c_0,c_1} \hat{\boldsymbol{A}}_{c_1,c_2} \cdots \hat{\boldsymbol{A}}_{c_{l-1},c_l} \boldsymbol{X}^{c_l} & l > 0 \end{cases}, \tag{1}$$

where $\boldsymbol{X}_k$ is the feature matrices of meta-path $P_k$, and $c_0$ is the target node type. Then, an MLP-based feature projection block is used to project different feature matrices into the same dimension, namely, $\boldsymbol{X}'_k = \text{MLP}_k(\boldsymbol{X}_k)$.

To automatically discover meaningful meta-paths for various datasets or tasks without prior, our search space contains all target-node-related meta-paths, severely challenging the efficiency and effectiveness of the search. To address the efficiency challenge, LMSPS utilizes a progressive sampling algorithm to sample meta-paths in each iteration and progressively shrink the search space.

Specifically, LMSPS assigns one architecture parameter to each meta-path. Let $\boldsymbol{\alpha} = \{\alpha_1, \cdots, \alpha_k, \cdots, \alpha_K\} \in \mathbb{R}^K$ be the architecture parameters of $K$ meta-paths. We use a Gumbel-softmax [35, 9] over architecture parameters $\alpha_k$ to calculate the strength of different meta-paths:

$$q_k = \frac{\exp\left[(\alpha_k + u_k)/\tau\right]}{\sum_{j=1}^K \exp\left[(\alpha_j + u_j)/\tau\right]}, \tag{2}$$

where $q_k$ is the path strength, representing the relative importance of meta path $P_k$. $u_k = -\log(-\log(U))$ where $U \sim \text{Uniform}(0,1)$, and $\tau$ is temperature controlling relaxation's extent.

The progressive sampling algorithm uses the path strength to progressively narrow the search space from $K$ to $V$ to exclude useless meta-paths. Generally, $K \gg V$ under a large maximum hop. Let $\tilde{q}_C$ be the $C$-th largest path strength of $\mathbb{Q} = \{q_1, \cdots, q_k, \cdots, q_K\}$. During the search stage, the search space retains all meta-paths no less than $\tilde{q}_C$. The dynamic search space can be formulated as follows:

$$\mathbb{S}_C = \{k | q_k \geq \tilde{q}_C, \forall 1 \leq k \leq K\} \quad \text{where} \quad C = \lceil \lambda(K - V) \rceil + V. \tag{3}$$

Here $C$ is the search space size, $\mathbb{S}_C$ consists of the indexes of retained meta-paths, and $\lceil \cdot \rceil$ indicates the rounding symbol. $\lambda \in [0, 1]$ is a parameter controlling the number of retrained meta-paths and decreases from 1 to 0 as the epoch increases.

As the search stage aims to determine top-$M$ meta-paths, we sample $M$ meta-paths from dynamic search space in each iteration. In each iteration, only parameters on the $M$ activated meta-paths are updated, while others remain unchanged. Therefore, the search cost is relevant to $M$ instead of $K$. The forward propagation can be expressed as:

$$\boldsymbol{Z} = \text{MLP}\Big( \sum_{k \in \mathbb{S}} q'_k \cdot \text{MLP}_k(\boldsymbol{X}_k) \Big) \quad \text{where} \ \mathbb{S} = \text{UniformSample}(\mathbb{S}_C, M). \tag{4}$$

Here $q'_k = \exp\left[(\alpha_k + u_k)/\tau\right] / \sum_{k \in \mathbb{S}} \exp\left[(\alpha_j + u_j)/\tau\right]$ indicates the path strength of activated meta-paths, and $\text{UniformSample}(\mathbb{S}_C, M)$ indicates a set of $M$ elements chosen randomly from set $\mathbb{S}_C$ without replacement via a uniform distribution.

The parameter update in the super-net involves a bilevel optimization problem [2, 7, 51].

$$\min_{\boldsymbol{\alpha}} \ \mathcal{L}_{val}(\mathbb{S}, \boldsymbol{\omega}^*(\boldsymbol{\alpha}), \boldsymbol{\alpha}) \quad \text{s.t.} \ \boldsymbol{\omega}^*(\boldsymbol{\alpha}) = \text{argmin}_{\boldsymbol{\omega}} \mathcal{L}_{train}(\mathbb{S}, \boldsymbol{\omega}, \boldsymbol{\alpha}). \tag{5}$$

Here $\mathcal{L}_{train}$ and $\mathcal{L}_{val}$ denote the training and validation loss, respectively. $\boldsymbol{\alpha}$ is the architecture parameters calculating the path strength. $\boldsymbol{\omega}$ is the network weights in MLP. Following the NAS-related works in the computer vision field [33, 50, 53], we address this issue by first-order approximation. Specifically, we alternatively freeze architecture parameters $\boldsymbol{\alpha}$ when training $\boldsymbol{\omega}$ on the training set and freeze $\boldsymbol{\omega}$ when training $\boldsymbol{\alpha}$ on the validation set.

The progressive sampling strategy can contribute to a more compact search space specifically driven by the current HIN and task, leading to a more effective meta-path discovery. Additionally, it can overcome the deep coupling issue [13] because of the randomness in each iteration.

## 5.2 Sampling Evaluation

After the completion of the progressive sampling search, the search space is narrowed from $K$ to $V$. Traditional methods in the computer vision field directly derive the final architecture based on the architecture parameters [33, 50, 53]. However, as different meta-paths could be noisy or redundant to each other, top-$M$ meta-paths are not necessarily the optimal solution when their importance is calculated independently. Based on this consideration, we specially designed a novel sampling evaluation strategy by evaluating the overall performance of each meta-path set. Specifically, using path strength at the end of progressive sampling as the probability, we sample $M$ meta-paths from the compact search space $\mathbb{S}_V$ to evaluate their overall performance. The sampling evaluation is repeated 200 times to filter out the meta-path set with the lowest validation loss. So, we can select the best meta-path set instead of independent top-$M$ meta-paths, which is more reasonable. This stage is not time-consuming because the evaluation does not involve weight training. This sampling process can be represented as:

$$\bar{\mathbb{S}} = \text{DiscreteSample}(\mathbb{S}_V, M, \bar{\mathbb{Q}}). \tag{6}$$

Here, $\text{DiscreteSample}(\mathbb{S}_V, M, \bar{\mathbb{Q}})$ indicates a set of $M$ elements chosen from set $\mathbb{S}_V$ without replacement via discrete probability distribution $\bar{\mathbb{Q}}$. $\bar{\mathbb{Q}}$ is the set of relative path strength calculated by architecture parameters of the $V$ meta-paths based on Equation 2. The overall search algorithm and more discussion are shown in Appendix C.

Thereafter, the retained meta-path set is recorded as $\mathbb{S}_M = \text{argmin}_{\bar{\mathbb{S}}} \mathcal{L}_{val}(\bar{\mathbb{S}}, \boldsymbol{\omega}^*, \boldsymbol{\alpha}^*)$. The forward propagation of the target-net for representation learning can be formulated as:

$$\hat{\boldsymbol{Z}} = \text{MLP}\Big( \big\|_{k \in \mathbb{S}_M} \text{MLP}_k(\boldsymbol{X}_k) \Big). \tag{7}$$

Here, $\big\|$ denotes the concatenation operation. Unlike existing HGNNs, the architecture of the target-net does not contain neighbor attention and semantic attention. Instead, the parametric modules consist of pure MLPs. The training objective is shown in Appendix A.3.

## 5.3 Discussion on Differences with Prior Works

As discussed in Section 3, there have been some attempts to find meta-paths automatedly [15, 56, 60]. Three aspects highlight the differences between LMSPS and existing methods. 1) Large-scale dataset: LMSPS is the first HGNN that makes it possible to achieve automated meta-path selection for large-scale heterogeneous graph node property prediction. 2) Long-range dependency: LMSPS is

Table 1: Performance on small and large datasets. Best is in bold, and the runner-up is underlined.

| Method | DBLP | | IMDB | | ACM | | Freebase | | OGBN-MAG* |
|---|---|---|---|---|---|---|---|---|---|
| | Macro-F1 | Micro-F1 | Macro-F1 | Micro-F1 | Macro-F1 | Micro-F1 | Macro-F1 | Micro-F1 | Test Acc. |
| MLP [19] | - | - | - | - | - | - | - | - | $26.92 \pm 0.26$ |
| GraphSAGE [14] | - | - | - | - | - | - | - | - | $46.78 \pm 0.67$ |
| RGCN [41] | $91.52 \pm 0.50$ | $92.07 \pm 0.50$ | $58.85 \pm 0.26$ | $62.05 \pm 0.15$ | $91.55 \pm 0.74$ | $91.41 \pm 0.75$ | $46.78 \pm 0.77$ | $58.33 \pm 1.57$ | $47.37 \pm 0.48$ |
| HAN [48] | $91.67 \pm 0.49$ | $92.05 \pm 0.62$ | $57.74 \pm 0.96$ | $64.63 \pm 0.58$ | $90.89 \pm 0.43$ | $90.79 \pm 0.43$ | $21.31 \pm 1.68$ | $54.77 \pm 1.40$ | $OOM$ |
| GTN [56] | $93.52 \pm 0.55$ | $93.97 \pm 0.54$ | $60.47 \pm 0.98$ | $65.14 \pm 0.45$ | $91.31 \pm 0.70$ | $91.20 \pm 0.71$ | $OOM$ | $OOM$ | $OOM$ |
| RSHN [63] | $93.34 \pm 0.58$ | $93.81 \pm 0.55$ | $59.85 \pm 3.21$ | $64.22 \pm 1.03$ | $90.50 \pm 1.51$ | $90.32 \pm 1.54$ | $OOM$ | $OOM$ | $OOM$ |
| HetGNN [57] | $91.76 \pm 0.43$ | $92.33 \pm 0.41$ | $48.25 \pm 0.67$ | $51.16 \pm 0.65$ | $85.91 \pm 0.25$ | $86.05 \pm 0.25$ | $OOM$ | $OOM$ | $OOM$ |
| MAGNN [11] | $93.28 \pm 0.51$ | $93.76 \pm 0.45$ | $56.49 \pm 3.20$ | $64.67 \pm 1.67$ | $90.88 \pm 0.64$ | $90.77 \pm 0.65$ | $OOM$ | $OOM$ | $OOM$ |
| HetSANN [18] | $78.55 \pm 2.42$ | $80.56 \pm 1.50$ | $49.47 \pm 1.21$ | $57.68 \pm 0.44$ | $90.02 \pm 0.35$ | $89.91 \pm 0.37$ | $OOM$ | $OOM$ | $OOM$ |
| GCN [26] | $90.84 \pm 0.32$ | $91.47 \pm 0.34$ | $57.88 \pm 1.18$ | $64.82 \pm 0.64$ | $92.17 \pm 0.24$ | $92.12 \pm 0.23$ | $27.84 \pm 3.13$ | $60.23 \pm 0.92$ | $OOM$ |
| GAT [46] | $93.83 \pm 0.27$ | $93.39 \pm 0.30$ | $58.94 \pm 1.35$ | $64.86 \pm 0.43$ | $92.26 \pm 0.94$ | $92.19 \pm 0.93$ | $40.74 \pm 2.58$ | $65.26 \pm 0.80$ | $OOM$ |
| Simple-HGN [34] | $94.01 \pm 0.24$ | $94.46 \pm 0.22$ | $63.53 \pm 1.36$ | $67.36 \pm 0.57$ | $93.42 \pm 0.44$ | $93.35 \pm 0.45$ | $47.72 \pm 1.48$ | $66.29 \pm 0.45$ | $OOM$ |
| HGT [21] | $93.01 \pm 0.23$ | $93.49 \pm 0.25$ | $63.00 \pm 1.19$ | $67.20 \pm 0.57$ | $91.12 \pm 0.76$ | $91.00 \pm 0.76$ | $29.28 \pm 2.52$ | $60.51 \pm 1.16$ | $46.78 \pm 0.42$ |
| GraphMSE [31] | $94.08 \pm 0.14$ | $94.44 \pm 0.13$ | $57.60 \pm 2.13$ | $62.37 \pm 1.03$ | $92.58 \pm 0.50$ | $92.54 \pm 0.14$ | $OOM$ | $OOM$ | $OOM$ |
| DiffMG [8] | $94.01 \pm 0.37$ | $94.20 \pm 0.36$ | $58.09 \pm 1.35$ | $59.75 \pm 1.23$ | $88.16 \pm 2.83$ | $88.07 \pm 3.04$ | $OOM$ | $OOM$ | $OOM$ |
| *Random* | $93.57 \pm 0.64$ | $93.84 \pm 0.53$ | $52.13 \pm 0.74$ | $53.83 \pm 0.66$ | $90.91 \pm 1.02$ | $90.82 \pm 0.93$ | $21.22 \pm 2.58$ | $37.54 \pm 2.66$ | $35.14 \pm 3.78$ |
| NARS [55] | $94.18 \pm 0.47$ | $94.61 \pm 0.42$ | $63.51 \pm 0.68$ | $66.18 \pm 0.70$ | $93.36 \pm 0.32$ | $93.31 \pm 0.33$ | $49.98 \pm 1.77$ | $63.26 \pm 1.26$ | $50.66 \pm 0.22$ |
| space4HGNN [59] | $94.24 \pm 0.42$ | $94.63 \pm 0.40$ | $61.57 \pm 1.19$ | $63.96 \pm 0.43$ | $92.50 \pm 0.14$ | $92.38 \pm 0.10$ | $41.37 \pm 4.49$ | $65.66 \pm 4.94$ | $OOM$ |
| PMMM [27] | $94.82 \pm 0.26$ | $95.14 \pm 0.22$ | $65.81 \pm 0.29$ | $67.58 \pm 0.22$ | $93.78 \pm 0.25$ | $93.71 \pm 0.17$ | $OOM$ | $OOM$ | $OOM$ |
| HINormer [36] | $94.57 \pm 0.23$ | $94.94 \pm 0.21$ | $64.65 \pm 0.53$ | $67.83 \pm 0.34$ | $93.91 \pm 0.42$ | $93.83 \pm 0.45$ | $\underline{52.18 \pm 0.39}$ | $64.92 \pm 0.43$ | $OOM$ |
| SeHGNN [52] | $94.86 \pm 0.14$ | $95.24 \pm 0.13$ | $\underline{66.63 \pm 0.34}$ | $68.21 \pm 0.32$ | $93.95 \pm 0.48$ | $93.87 \pm 0.50$ | $50.71 \pm 0.44$ | $63.41 \pm 0.47$ | $\underline{51.45 \pm 0.29}$ |
| SlotGAT [62] | $\underline{94.95 \pm 0.20}$ | $\underline{95.31 \pm 0.19}$ | $64.05 \pm 0.60$ | $\underline{68.64 \pm 0.33}$ | $\underline{93.99 \pm 0.23}$ | $\underline{94.06 \pm 0.22}$ | $49.68 \pm 1.97$ | $\mathbf{66.83 \pm 0.30}$ | $OOM$ |
| LMSPS (ours) | $\mathbf{95.35 \pm 0.22}$ | $\mathbf{95.66 \pm 0.20}$ | $\mathbf{66.99 \pm 0.32}$ | $\mathbf{68.70 \pm 0.26}$ | $\mathbf{94.73 \pm 0.41}$ | $\mathbf{94.69 \pm 0.36}$ | $\mathbf{53.26 \pm 0.47}$ | $66.09 \pm 0.51$ | $\mathbf{54.83 \pm 0.20}$ |

$^*$ OGBN-MAG is a large dataset with nodes' numbers 10 to 175 times that of the other four datasets.

the first HGNN to utilize long-range dependency in large-scale heterogeneous graphs. To achieve the above two goals, LMSPS has addressed two key challenges: (a) alleviating costs while striving to effectively utilize information in exponentially increased receptive fields and (b) overcoming the well-known over-smoothing issue. 3) High generalization: Based on Table 4, the searched meta-paths of LMSPS can be generalized to other HGNNs to boost their performance, which has not been achieved by existing works. To accomplish this objective, LMSPS uses an MLP-based architecture instead of a transformer-based one for meta-path search because the former involves fewer inductive biases [45, 16, 3], i.e., human interventions.

## 6  Experiments and Analysis

This section evaluates the benefits of our method against state-of-the-art models on nine heterogeneous datasets. We aim to answer the following questions: **Q1.** How does LMSPS perform compared with state-of-the-art baselines? **Q2.** Can LMSPS overcome the over-smoothing and noise issues? **Q3.** Are the search algorithm and searched meta-paths effective? **Q4.** Does LMSPS perform better on sparser heterogeneous graphs?

### 6.1  Datasets and Baselines

We evaluate LMSPS on several representative heterogeneous graph datasets, including `DBLP`, `IMDB`, `ACM`, and `Freebase` from HGB benchmark [34], and the large-scale dataset `OGBN-MAG` from OGB challenge [20]. The statistics of the datasets are listed in Table 7. The details about all datasets, baselines, and experiment settings are recorded in Appendix A and B.

### 6.2  Performance Analysis

To answer **Q1**, we report the performance of our approaches and baselines in Table 1. *Random* means the result of replacing our searched meta-paths with 30 random meta-paths. We show the average result of 20 random samples. Based on the results, we have the following observations. First, LMSPS outperforms all baselines for different metrics on almost all datasets except Micro-F1 scores on Freebase, sometimes by a significant margin, which validates the power of LMSPS. For example, on the largest dataset `OGBN-MAG`, LMSPS achieves $54.83\%$ test accuracy, while the best competitor has $51.45\%$ test accuracy. Second, all metapath-free methods encounter out-of-memory (OOM) issues when dealing with large datasets, including highly competitive methods, HINormer and SlotGAT, indicating the advantage of employing meta-paths electively aggregating neighbors on the meta-path instances. Third, although LMSPS is an MLP-based method, the pure MLP method [19] has the worst performance with only $26.92\%$ test accuracy on `OGBN-MAG`, validating the advantages of pre-processing and meta-path search. Finally, LMSPS outperforms *Random* significantly, further indicating the importance of meta-path search. More comparisons with top method combinations

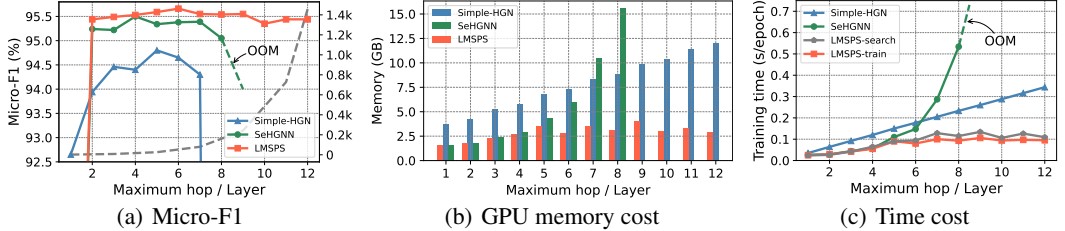

(a) Micro-F1      (b) GPU memory cost      (c) Time cost

Figure 3: Illustration of (a) performance, (b) memory cost, (c) average training time of Simple-HGN, SeHGNN, and LMSPS relative to the maximum hop or layer on DBLP. The *gray dotted line* in (a) indicates the number of target-node-related meta-paths under different maximum hops, which is exponential.

Table 2: Experiments on OGBN-MAG to analyze the performance of SeHGNN and LMSPS under different maximum hops. #MP is the number of meta-paths under different maximum hops.

| Max hop | #MP | SeHGNN | | LMSPS | |
|---|---|---|---|---|---|
| | | Time | Test accuracy | Time | Test accuracy |
| 1 | 4 | 4.35 | $47.18 \pm 0.28$ | 3.98 | $46.88 \pm 0.10$ |
| 2 | 10 | 6.44 | $51.79 \pm 0.24$ | 5.63 | $51.91 \pm 0.13$ |
| 3 | 23 | 11.28 | $52.44 \pm 0.16$ | 10.02 | $52.72 \pm 0.24$ |
| 4 | 50 | OOM | OOM | 14.34 | $53.43 \pm 0.18$ |
| 5 | 107 | OOM | OOM | 14.77 | $53.90 \pm 0.19$ |
| 6 | 226 | OOM | OOM | 14.71 | $\mathbf{54.83 \pm 0.20}$ |

Table 3: Experiments to explore the effectiveness of our search algorithm. In our LMSPS, the meta-paths are replaced by those discovered by other methods.

| Method | DBLP | IMDB | ACM | Freebase |
|---|---|---|---|---|
| HAN | $95.44 \pm 0.14$ | $65.95 \pm 0.31$ | $90.66 \pm 0.30$ | - |
| GTN | $95.33 \pm 0.05$ | $65.99 \pm 0.16$ | $90.66 \pm 0.30$ | - |
| DARTS | $95.35 \pm 0.17$ | $66.23 \pm 0.14$ | $93.45 \pm 0.13$ | $63.25 \pm 0.42$ |
| SPOS | $95.41 \pm 0.43$ | $67.10 \pm 0.29$ | $93.64 \pm 0.37$ | $64.02 \pm 0.62$ |
| DiffMG | $95.45 \pm 0.49$ | $66.98 \pm 0.37$ | $93.61 \pm 0.45$ | $64.56 \pm 0.78$ |
| PMMM | $95.48 \pm 0.27$ | $67.49 \pm 0.24$ | $93.74 \pm 0.22$ | $64.83 \pm 0.46$ |
| LMSPS | $\mathbf{95.66 \pm 0.20}$ | $\mathbf{68.70 \pm 0.26}$ | $\mathbf{94.69 \pm 0.36}$ | $\mathbf{66.09 \pm 0.51}$ |

on OGBN-MAG are shown in Table 10 of Appendix E.2, in which LMSPS also shows the best performance.

## 6.3 Analysis on Large Maximum Hops

To answer **Q2**, we conduct experiments to compare the performance, memory, and efficiency of LMSPS with the method of HGB benchmark, Simple-HGN [34], and the best metapath-based method, SeHGNN [52], on DBLP. Figure 3 (a) shows that LMSPS has consistent performance with the increment of maximum hop. The failure of Simple-HGN demonstrates its attention mechanism can not overcome the over-smoothing issue or eliminate the effects of noise under large hop. Figure 3 (b), (c) illustrate each training epoch's average memory and time costs relative to the maximum hop or layer. We can observe that the consumption of SeHGNN exponentially grows, and the consumption of Simple-HGN linearly increases, which is consistent with their time complexity as listed in Table 8. Meanwhile, LMSPS has almost constant consumption as the maximum hop grows. Figure 3 (c) shows the time cost of LMSPS in the search stage, which also approximates a constant when the number of meta-paths is larger than $M = 30$. More results about efficiency are shown in Figure 4 of Appendix E.1.

We also compare the performance and training time of LMSPS with SeHGNN under different maximum hops on large-scale dataset OGBN-MAG. When the maximum hop $l = 1, 2, 3$, we utilize the full meta-path set because the number of meta-paths is smaller than $M = 30$. Following the convention [34, 52], we measure the average time consumption of one epoch for each model. As shown in Table 2, the performance of LMSPS keeps increasing as the maximum hop value grows. It indicates that LMSPS can overcome the issues caused by utilizing long-range dependency, *e.g.*, over-smoothing and noise. In addition, when the number of meta-paths is larger than 30, the training time of LMSPS is stable under different maximum hops.

## 6.4 Effectiveness of the Search Algorithm and Searched Meta-paths

To answer **Q3**, we first explore the effectiveness of our search algorithm. In our architecture, our meta-paths are replaced by those meta-paths discovered by other methods. DARTS [32] is the first differentiable search algorithm in neural networks. SPOS [13] is a classic singe-path differentiable algorithm. DARTS and SPOS aim to search operations, like $3 \times 3$ convolution, in CNNs. DiffMG [8] and PMMM [27] search for meta-graphs instead of meta-paths. We ignore these differences and focus on the algorithms. The derivation strategies of the four methods are unsuitable for discovering

Table 4: Experiments on the generalization of the searched meta paths. * means using the meta-paths searched in LMSPS.

| Method | DBLP | IMDB | ACM | Freebase |
|---|---|---|---|---|
| HAN | $92.05 \pm 0.62$ | $64.63 \pm 0.58$ | $90.79 \pm 0.43$ | $54.77 \pm 1.40$ |
| HAN* | $93.54 \pm 0.15$ | $65.89 \pm 0.52$ | $92.28 \pm 0.47$ | $57.13 \pm 0.72$ |
| SeHGNN | $95.24 \pm 0.13$ | $68.21 \pm 0.32$ | $93.87 \pm 0.50$ | $63.41 \pm 0.47$ |
| SeHGNN* | $95.57 \pm 0.23$ | $68.59 \pm 0.24$ | $94.46 \pm 0.18$ | $65.37 \pm 0.42$ |

Table 5: Results of LMSPS and SeHGNN on the sparse large-scale heterogeneous graphs. $\uparrow$ means the improvements in test accuracy.

| Dataset | SeHGNN | LMSPS | $\uparrow$ |
|---|---|---|---|
| OGBN-MAG-5 | $36.04 \pm 0.64$ | $40.82 \pm 0.42$ | **4.78** |
| OGBN-MAG-10 | $38.27 \pm 0.19$ | $42.30 \pm 0.23$ | **4.03** |
| OGBN-MAG-20 | $39.18 \pm 0.09$ | $42.65 \pm 0.17$ | **3.47** |
| OGBN-MAG-50 | $39.50 \pm 0.13$ | $42.82 \pm 0.16$ | **3.32** |

Table 6: Experiments on small and large datasets to analyze the effects of different blocks in LMSPS. *PS* means progressive sampling strategy, and *SE* means sampling evaluation strategy. $\dagger$ means employing all meta-paths and replacing the concatenation operation with the transformer module.

| Method | DBLP | | IMDB | | ACM | | Freebase | | OGBN-MAG |
|---|---|---|---|---|---|---|---|---|---|
| | Macro-F1 | Micro-F1 | Macro-F1 | Micro-F1 | Macro-F1 | Micro-F1 | Macro-F1 | Micro-F1 | Test Acc. |
| LMSPS w/o *PS* | $94.71 \pm 0.23$ | $95.00 \pm 0.19$ | $64.85 \pm 0.46$ | $66.52 \pm 0.37$ | $93.19 \pm 0.34$ | $93.14 \pm 0.41$ | $48.89 \pm 0.47$ | $61.61 \pm 0.51$ | $47.66 \pm 0.45$ |
| LMSPS w/o *SE* | $95.15 \pm 0.28$ | $95.48 \pm 0.24$ | $65.46 \pm 0.48$ | $67.13 \pm 0.47$ | $94.20 \pm 0.35$ | $94.15 \pm 0.31$ | $52.08 \pm 0.33$ | $64.84 \pm 0.38$ | $52.94 \pm 0.34$ |
| LMSPS $\dagger$ | $95.06 \pm 0.24$ | $95.38 \pm 0.21$ | $66.85 \pm 0.37$ | $68.58 \pm 0.34$ | $94.60 \pm 0.42$ | $94.57 \pm 0.39$ | *OOM* | *OOM* | *OOM* |
| LMSPS | **$95.35 \pm 0.22$** | **$95.66 \pm 0.20$** | **$66.99 \pm 0.32$** | **$68.70 \pm 0.26$** | **$94.73 \pm 0.41$** | **$94.69 \pm 0.36$** | **$53.26 \pm 0.47$** | **$66.09 \pm 0.51$** | **$54.83 \pm 0.20$** |

multiple meta-paths. So we changed their derivation strategies to ours to improve their performance. we report the Micro-F1 scores in Table 3. The performance of LMSPS verifies the effectiveness of our search algorithm.

To demonstrate the effectiveness of searched meta-paths, on the one hand, the meta-paths should be effective in the proposed model. On the other hand, the effective meta-paths mainly depend on the dataset instead of the architecture, so the meta-paths should be effective after being generalized to other HGNNs. Because finding meta-paths that work effectively across various HGNNs is a tough task, it has not been achieved by previous works. We verify the generalization of our searched meta-path on the most famous HGNN, HAN [48], and the SOTA metapath-based method SeHGNN [52]. The Micro-F1 scores on three representative datasets are shown in Table 4. After simply replacing the original meta-path set with our searched meta-paths and keeping other settings unchanged, the performance both improve, demonstrating the effectiveness of our searched meta-paths. In addition, we have shown the interpretability of searched meta-paths in Appendix D.

## 6.5 Necessity of Long-range Dependency

To answer **Q4** and explore the necessity of long-range dependency on heterogeneous graphs, we construct four large-scale datasets with high sparsity based on `OGBN-MAG`. To avoid inappropriate preference seed settings of randomly removing, we construct fixed heterogeneous graphs by limiting the maximum in-degree related to edge type. Specifically, we gradually reduce the maximum in-degree related to edge type in `OGBN-MAG` from $50$ to $5$ but leave all nodes unchanged. Details of the four datasets are listed in Appendix A. The test accuracy of LMSPS and SOTA method SeHGNN are shown in Table 5. LMSPS outperforms SeHGNN more significantly with the increasing sparsity. In addition, the leading gap of LMSPS over SeHGNN is more than $4.7\%$ on the highly sparse dataset `OGBN-MAG-5`. The main difference between SeHGNN and LMSPS is that the former cannot utilize large hops and only use hop 2 while the latter has a maximum hop of 6, demonstrating that long-range dependencies are more effective with decreased direct neighbors on heterogeneous graphs.

## 6.6 Ablation Study

Two components differentiate our LMSPS from other HGNNs: the search algorithm and semantic fusion without attention. The search algorithm consists of a progressive sampling algorithm and a sampling evaluation strategy. We explore how each of them improves performance through ablation studies under the same settings as the main experiments in Table 1. As shown in Table 6, the performance of LMSPS significantly decreases when removing progressive sampling or sampling evaluation strategy. In addition, employing a transformer block for semantic attention on all meta-paths shows slightly worse performance even if using many more meta-paths and is out-of-memory on larger datasets, indicating that the transformer cannot eliminate the negative effects of noise. It is reasonable because attention values are calculated based on softmax and are positive even for noise.

# 7 Conclusion

This work presented a novel approach, the Long-range Meta-path Search through Progressive Sampling (LMSPS), to tackle the challenges of leveraging long-range dependencies in large-scale heterogeneous graphs, *i.e.*, reducing computational costs while effectively utilizing information and addressing the over-smoothing issue. Based on our two observations, *i.e.*, a few meta-paths dominate the performance, and certain meta-paths can have negative impact on performance, LMSPS introduced a progressive sampling search algorithm and a sampling evaluation strategy to automatically identify effective meta-paths, hence reducing the exponentially growing number of meta-paths to a manageable constant. Extensive experiments demonstrated the superiority of LMSPS over existing methods, particularly on sparse heterogeneous graphs that require long-range dependencies. By employing simple MLPs and complex meta-paths, LMSPS offers a novel direction that emphasizes data-dependent semantic relationships rather than relying solely on sophisticated neural architectures. The reproducibility and limitations are discussed in Appendix F and Appendix G, respectively.

## Acknowledgments

This work is supported by National Natural Science Foundation (62076105,U22B2017).

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

# Appendix / Supplemental Material

In the Appendix, we provide additional details and results not included in the main text due to space limitations. First, we provide the details of datasets, tasks, baselines, and parameter settings for our experiments. Secondly, we show the search algorithm of LMSPS and conduct theoretical analysis about the reasonableness of sampling search and analyze the time complexity. Thirdly, for interpretability, the searched meta-paths are listed and analyzed. Fourthly, we explore the training efficiency and search convergence of LMSPS and compare LMSPS with top method combinations on `OGBN-MAG` leaderboard. Hyperparameter studies are also conducted. Finally, we discuss the reproducibility and limitations of our research work.

## A    Dataset and Task

### A.1    Dataset Details

We evaluate our method on four widely-used heterogeneous graphs including `DBLP`, `IMDB`, `ACM`, and `Freebase` from HGB benchmark [34] and the famous large-scale dataset `OGBN-MAG` from OGB challenge [20]. The datasets from HGB follow a transductive setting, where all edges are available during training, and target type nodes are divided into $24\%$ for training, $6\%$ for validation, and $70\%$ for testing. For the `OGBN-MAG` dataset, we use the official data partition, where papers published before 2018, in 2018, and since 2019 are nodes for training, validation, and testing, respectively. In addition, we construct four sparse datasets to evaluate the performance of LMSPS by reducing the maximum in-degree related to edge type in `OGBN-MAG`. The four sparse datasets have the same number of nodes with `OGBN-MAG`. Follow the convention [34, 52, 36], we show the details of all datasets in Table 7.

### A.2    Node Classification Tasks

Following the convention [34, 27, 52, 36], we concentrate on semi-supervised node classification under the transductive setting and leave other downstream tasks related to heterogeneous graph representation learning as future work. Each dataset contains a target node type, and our task is to learn to predict the labels of vertices of this target node type. All the datasets provide fixed data splitting (into training, validation, and test sets) for node classification tasks.

### A.3    Searching and Training Objective

An MLP layer is appended after the main module of LMSPS, which reduces the dimension of node-level representations of a heterogeneous graph to the number of classes:

$$\bar{Y} = \text{softmax}(Z\omega_o), \tag{8}$$

where $\omega_o \in \mathbb{R}^{d_o \times N}$ is the output weight matrix, $d_o$ is the dimension of output node representations, and $N$ is the number of classes. Then, the cross-entropy loss is used over all labeled nodes:

$$\mathcal{L} = -\sum_{v \in \mathcal{V}_L} \sum_{n=1}^{N} \boldsymbol{y}_v[n] \log \bar{\boldsymbol{y}}_v[n], \tag{9}$$

Table 7: Statistics of the datasets.

| Dataset | #Nodes | #Node types | #Edges | #Edge types | Target | #Classes |
|---|---|---|---|---|---|---|
| DBLP | $26,128$ | 4 | $239,566$ | 6 | author | 4 |
| IMDB | $21,420$ | 4 | $86,642$ | 6 | movie | 5 |
| ACM | $10,942$ | 4 | $547,872$ | 8 | paper | 3 |
| Freebase | $180,098$ | 8 | $1,057,688$ | 36 | book | 7 |
| OGBN-MAG | $1,939,743$ | 4 | $21,111,007$ | 4 | paper | 349 |
| OGBN-MAG-50 | $1,939,743$ | 4 | $9,531,403$ | 4 | paper | 349 |
| OGBN-MAG-20 | $1,939,743$ | 4 | $7,975,003$ | 4 | paper | 349 |
| OGBN-MAG-10 | $1,939,743$ | 4 | $6,610,464$ | 4 | paper | 349 |
| OGBN-MAG-5 | $1,939,743$ | 4 | $4,958,941$ | 4 | paper | 349 |

where $\mathcal{V}_L$ denotes the set of labeled nodes, $\boldsymbol{y}_v$ is a one-hot vector indicating the label of node $v$, and $\bar{\boldsymbol{y}}_v$ is the predicted label for the corresponding node in $\bar{Y}$. In the search stage, $\mathcal{V}_L$ is the training set when updating $\omega$ and the validation set when updating $\alpha$. In the training stage, $\mathcal{V}_L$ is the training set, and $\omega$ is reinitialized and trained.

## B  Baselines and Parameter Settings

### B.1  Baselines

We compare LMSPS with a large number of HGNN baselines, including MLP [19], GraphSAGE [14], RGCN [41], HAN [48], GTN [56], RSHN [63], HetGNN [57], MAGNN [11], HetSANN [18], GCN [26], GAT [46], , Simple-HGN [34], HGT [21], GraphMSE [31], SlotGAT [62], DiffMG [8], PMMM [27], HINormer [36], NARS [55], space4HGNN [59], SeHGNN [52]. GEMS [15] and RMS-HRec [37] are ignored because they are designed for recommendation tasks. RL-HGNN [60] is omitted due to a lack of source code.

Note that most of these baselines encounter out-of-memory (OOM) issues when dealing with large datasets. However, there are exceptions among them, including MLP, GraphSAGE, HGT, RGCN, NARS, and SeHGNN. The results of MLP and GraphSAGE come from the OGBN-MAG leaderboard. MLP [19] is famous for its lightweight and GraphSAGE [14] uses a neighbor sampling strategy, so they can handle large-scale datasets. HGT is designed with a graph sampling strategy, making it well-suited for managing large datasets. As for RGCN, while the original version does present OOM issues with large datasets, it is often used as a baseline for large-scale HGNNs [55]. NARS [55] and SeHGNN [52] are metapath-based and pre-computation-based HGNNs, allowing them to handle large-scale datasets.

It's worth noting that in the original paper, SeHGNN refers to the full version incorporating label propagation techniques. For clarity and consistency in our study, we have adjusted the terminology: we refer to the version without label propagation as SeHGNN and the version with label propagation as SeHGNN+LP. Note that label propagation is a general technique [64, 61, 22] that is compatible with most GNNs and is not used in the results of LMSPS and all other baselines in Table 1. Therefore, it is considered only in Appendix E.2, where we demonstrate the successful integration of our model with other techniques.

### B.2  Training Settings

For training, we follow the settings of the HGB benchmark [34] and utilize the performance improvement on the validation set as a guide to determine whether the model has been improved. Specifically, if we observe a performance boost on the validation set during training, we update the final model parameters accordingly. Additionally, we adopt the early stopping strategy employed by the HGB benchmark: if no performance improvement on the validation set is observed within a specific count (referred to as patience), the training will be stopped early before reaching the maximum epochs. This approach helps prevent overfitting while also enhancing the computational efficiency of our model training. Following the setting of HGB [34] and OGB [20], we train our model 5 times for HGB and 10 times for OGB with different random seeds and report the mean performance and standard deviation, respectively. We use Pytorch [38] to run all experiments on one Tesla V100 GPU with 16GB GPU memory.

### B.3  Parameter Settings for Baselines

For models such as RGCN, HAN, GTN, RSHN, HetGNN, MAGNN, HetSANN, HGT, GCN, GAT, Simple-HGN, and SeHGNN, the literature [34, 52] provides tuned parameter settings along with corresponding performance on small datasets, namely DBLP, IMDB, ACM, and Freebase. Therefore, we utilize these tuned parameter settings for these models on small datasets and utilize the reported performance.

Conversely, in cases where official results are unavailable (such as NARS on small datasets), or where the experimental settings differ, we employ the official or benchmark implementations [3] [4] [5] [6] [7] [8] [9] [10] [11] of the baseline models. We meticulously fine-tune their hyperparameters to the best of our capabilities. In instances involving hyperparameter analysis, such as in Figure 3, we only modify the relevant hyperparameters to ensure a fair comparison.

### B.4 Parameter Settings for LMSPS

We set the number of selected meta-paths $M = 30$ for all datasets. The final search space $V = 60$. The maximum hop is 6 for `ogbn-mag`, DBLP, 5 for IMDB, ACM, and 3 for `Freebase`. All $K$ architecture parameters $\alpha_1, \alpha_2, \cdots, \alpha_K$ are initialized as 1s. For searching in the super-net, we train for 200 epochs. To train the target-net, we use an early stop mechanism based on the validation performance to promise full training. A two-layer MLP is adopted for each meta-path in the feature projection step, and the hidden size is $512$. All network weights are initialized by the Xavier uniform distribution [12] and are optimized with Adam [25] during training. In the search stage, $\lambda$ is 1 during the first 20 epochs for warmup and decreases to 0 linearly. $\tau$ linearly decays with the number of epochs from 8 to 4. The learning rate is $0.001$ for all search stages and HGB training stage, and $0.003$ for `OGBN-MAG` training stage. The weight decay is always $0$. For the initial search space, we simply preset the maximum hop and use all target-node-related meta-paths no more than this maximum hop.

### B.5 Evaluation Metrics

To assess the performance of the models, we employ evaluation metrics consistent with those employed in baseline models. The metrics are chosen as follows. For small datasets, DBLP, IMDB, ACM, and Freebase, we adhere to the evaluation standards established by the HGB benchmark [34]. The metrics reported for these datasets are Macro-F1 and Micro-F1 scores, which evaluate the classification performance. For the OGBN-MAG dataset, evaluation follows the methodology outlined in NARS [55] and SeHGNN [52], where the classification accuracy score is reported for this dataset.

## C  Algorithm

### C.1  The Search Algorithm

Our search stage aims to discover the most effective meta-path set from all target-node-related meta-paths, severely challenging the efficiency of searching. Take `OGBN-MAG` as an example. The number of target-node-related meta-paths $K$ is 226 under the maximum hop 6, and we need to find the most effective meta-path set with size 30. Because different meta-paths could be noisy or redundant to each other, top-30 meta-paths are not necessarily the optimal solution when their importance is calculated independently. Therefore, the total number of possible meta-path sets is $C_{226}^{30} \approx 10^{37}$. Such a large search space is hard to solve efficiently by traditional RL-based algorithms [60, 37] or evolution-based algorithms [49, 39].

To overcome this challenge, our LMSPS first uses a progressive sample algorithm to shrink the search space size from 226 to 60, then utilizes a sampling evaluation strategy to discover the best meta-path set with the lowest validation loss, which is more effective than architecture parameters [47]. In each iteration, we only uniformly sample meta-paths $M$ from the whole search space for parameter updates, so the search cost is relevant to $M$, which is a predefined small number, rather than $K$. Because the search stage has many iterations and the initial values of architecture parameters are

---

[3] `https://github.com/THUDM/HGB`

[4] `https://github.com/dmlc/dgl/tree/master/examples/pytorch/ogb/ogbn-mag`

[5] `https://github.com/UCLA-DM/pyHGT`

[6] `https://github.com/facebookresearch/NARS`

[7] `https://github.com/LARS-research/DiffMG`

[8] `https://github.com/JHL-HUST/PMMM`

[9] `https://github.com/ICT-GIMLab/SeHGNN`

[10] `https://github.com/Ffffffffire/HINormer`

[11] `https://github.com/scottjiao/SlotGAT_ICML23`

---
**Algorithm 1** The search algorithm of LMSPS
---
**Input**: meta-path sets $\mathbb{P} = \{P_1, \cdots, P_K\}$; number of sampling meta-paths $M$; number of training iterations $T$; number of sampling evaluation $E$
**Parameter**: Network weights $\boldsymbol{\omega}$ in $\text{MLP}_k$ for feature projection and MLP for downstream tasks; architecture parameters $\boldsymbol{\alpha} = \{\alpha_1, \cdots, \alpha_K\}$
**Output**: The index set of selected meta-paths $\mathbb{S}_M$
---
 1: **% Neighbor aggregation**
 2: Calculate neighbor aggregation of raw features for each $P_k \in \mathbb{P}$ based on Equation 1
 3: **while** $t < T$ **do**
 4:     **% Path strength**
 5:     Calculate the path strength of all meta-paths based on Equation 2
 6:     **% Dynamic search space**
 7:     Calculate the current search space $\mathbb{S}_C$ based on Equations 3
 8:     **% Sampling**
 9:     Determine the set of indexes of sampled meta-paths $\mathbb{S}$ based on Equation 4
10:     **% Semantic fusion**
11:     Fused the semantic information of the sampled meta-paths based on Equation 4
12:     **% Parameters updation**
13:     Update weights $\boldsymbol{\omega}$ by $\nabla_\omega \mathcal{L}_{train}(\boldsymbol{\omega}, \boldsymbol{\alpha})$
14:     Update parameters $\boldsymbol{\alpha}$ by $\nabla_\alpha \mathcal{L}_{val}(\boldsymbol{\omega}, \boldsymbol{\alpha})$
15: **end while**
16: **% Evaluation**
17: **while** $e < E$ **do**
18:     Randomly sample $M$ meta-paths from $\mathbb{S}_C$ as $\bar{\mathbb{S}}$ based on Equation 6
19:     Calculate $\mathcal{L}_{val}(\bar{\mathbb{S}})$ of the sampled meta-paths
20: **end while**
21: Select the best meta-path set $\mathbb{S}_M \leftarrow \arg\min_{\bar{\mathbb{S}}} \mathcal{L}_{val}(\bar{\mathbb{S}})$
22: **return** $\mathbb{S}_M$
---

the same, all architecture parameters will be updated multiple times and the relative importance can be learned during training, making the total search cost similar to training a single HGNN once. Specifically, for `OGBN-MAG`, LMSPS can finish searching in two hours.

Except for efficiency, LMSPS can also overcome the over-squashing issue [1] when utilizing long-range dependency. Over-squashing means the distortion of messages being propagated from distant nodes, which has been heuristically attributed to graph bottlenecks where the number of $l$-hop neighbors grows rapidly with $l$. In LMSPS, we set the number of searched meta-paths $M = 30$ for all datasets, which is independent of $l$. Because the exponential meta-paths in metapath-based methods correspond to exponential receptive fields in metapath-free methods, the constant $M$ means approximately constant $l$-hop neighbors, *i.e.*, LMSPS selectively aggregate effective neighbors for each target-net. Therefore, the over-squashing issue is overcome.

We have introduced the components of LMSPS in detail in the main text. LMSPS first employs a progressive sample algorithm to narrow the search space from $K$ to $V$, then utilizes sample evaluation to filter out the best meta-path set with $M$ meta-paths. $V$ is a parameter to trade off the importance of progressive sampling search and sampling evaluation. When $V$ is too large, we need to repeat the sampling evaluation many more times, which will decrease the efficiency. When $V$ is too small, some effective meta-paths may be dropped too early. So, we simply set $V = 2M$. Generally, $K \gg M$ under a large maximum hop. For a small maximum hop, when $K \leq M$, the search stage is unnecessary because we can directly use all target-node-related meta-paths; when $K \leq 2M$, the progressive sampling algorithm is unnecessary because the search space is small enough. When $K > 2M$, we show the overall search algorithm in Algorithm 1.

### C.2 Theoretical analysis of the reasonableness of sampling search

**Definition 1.** ***Zero-order condition***: *Given two high-dimensional random variables,* $\mathbf{y} = \mathbf{f}(\mathbf{x}) \in \mathbf{R}^{\mathbf{M} \times \mathbf{d_1}}$ *and* $\mathbf{z} = \mathbf{g}(\mathbf{x}) \in \mathbf{R}^{\mathbf{M} \times \mathbf{d_1}}$, *the zero-order condition is satisfied if* $|\mathbf{y} - \mathbf{z}|_2 \leq \epsilon$ *for any valid sample* $\mathbf{x} \in \mathbf{R}^{\mathbf{N} \times \mathbf{d}}$, *where* $\epsilon$ *is a small positive constant.*

Table 8: Time complexity comparison of every training epoch. † means time complexity under small-scale datasets and full-batch training.

| Method | Feature projection | Neighbor aggregation | Semantic fusion | Total |
|--------|-------------------|---------------------|-----------------|-------|
| HAN | $O(N(rd)^l F^2)$ | $O(N(rd)^l F)$ | $O(Nr^l F^2)$ | $O(N(rd)^l F^2)$ |
| Simple-HGN | $O(N(rd)^l F^2)$ | $O(N(rd)^l F)$ | - | $O(N(rd)^l F^2)$ |
| Simple-HGN† | $O(NrdlF^2)$ | $O(NrdlF)$ | - | $O(NrdlF^2)$ |
| SeHGNN | $O(Nr^l F^2)$ | - | $O(Nr^l F^2 + Nr^{2l} F)$ | $O(N(r^l F^2 + r^{2l} F))$ |
| LMSPS-search | $O(NMF^2)$ | - | $O(NMF^2)$ | $O(NMF^2)$ |
| LMSPS-train | $O(NMF^2)$ | - | $O(NMF^2)$ | $O(NMF^2)$ |

**Lemma C.1.** *Let* $\mathbf{m}$ *represent the maximum number of activatable paths, and assume each pair of operations satisfies the zero-order condition. We can approximate all* $2^{\mathbf{m}}$ *combinations using* $\mathbf{m}$ *types of expectations and variances.*

Lemma C.1 assures us that we can approximate all $2^{\mathbf{m}}$ combinations by multiple times sampling. The proof is given as follows.

Let $\mathbf{y}_{p_{\mathbf{y}}(\mathbf{y})} = f(\mathbf{x})$, $\mathbf{z}_{p_{\mathbf{z}}(\mathbf{z})} = g(\mathbf{x})$, $\mathbf{x} \sim p_{\mathbf{x}}(\mathbf{x})$. For the case $m = 1$, the expectation of $\mathbf{y}$ and $\mathbf{z}$ can be written respectively as:

$$
\begin{aligned}
\mathbb{E}[\mathbf{y}] = \mathbb{E}[f(\mathbf{x})] = \int p_{\mathbf{x}}(\mathbf{x})f(\mathbf{x})\mathrm{d}\,\mathbf{x} \\
\mathbb{E}[\mathbf{z}] = \mathbb{E}[g(\mathbf{x})] = \int p_{\mathbf{x}}(\mathbf{x})g(\mathbf{x})\mathrm{d}\,\mathbf{x}
\end{aligned}
\tag{10}
$$

According to the zero-order condition, we have $f(\mathbf{x}) \approx g(\mathbf{x})$. And $p(\mathbf{x})$ is same for both $\mathbf{y}$ and $\mathbf{z}$, so $\mathbb{E}[\mathbf{y}] \approx \mathbb{E}[\mathbf{z}]$.

Now we prove $Var[\mathbf{y}] \approx Var[\mathbf{z}]$. Note that $Var[\mathbf{y}] = \mathbb{E}\left[\mathbf{y}^2\right] - \left(\mathbb{E}[\mathbf{y}]\right)^2$ and $Var[\mathbf{z}] = \mathbb{E}\left[\mathbf{z}^2\right] - \left(\mathbb{E}[\mathbf{z}]\right)^2$, thus we only need to prove $\mathbb{E}\left[\mathbf{y}^2\right] \approx \mathbb{E}\left[\mathbf{z}^2\right]$. It can be similarly proved as follows:

$$
\begin{aligned}
\mathbb{E}\left[\mathbf{y}^2\right] = \int p_{\mathbf{y}}(\mathbf{y})\mathbf{y}^2\mathrm{d}\mathbf{y} = \int p_{\mathbf{x}}(\mathbf{x})f^2(\mathbf{x})\mathrm{d}\mathbf{x} \\
\mathbb{E}\left[\mathbf{z}^2\right] = \int p_{\mathbf{z}}(\mathbf{z})\mathbf{z}^2\mathrm{d}\mathbf{z} = \int p_{\mathbf{x}}(\mathbf{x})g^2(\mathbf{x})\mathrm{d}\mathbf{x}
\end{aligned}
\tag{11}
$$

According to the zero-order condition, we have $Var[\mathbf{y}] \approx Var[\mathbf{z}]$.

For the case of $m = 2$, when the two paths are both selected, the output becomes $\mathbf{y} + \mathbf{z}$, its expectation can be written as:

$$
\mathbb{E}[\mathbf{y} + \mathbf{z}] = \mathbb{E}[\mathbf{y}] + \mathbb{E}[\mathbf{z}] \approx 2\mathbb{E}[\mathbf{y}]
\tag{12}
$$

And the variance of $\mathbf{y} + \mathbf{z}$ is,

$$
Var[\mathbf{y} + \mathbf{z}] \approx Var[2\mathbf{y}] = 4Var[\mathbf{y}]
\tag{13}
$$

Therefore, there are two kinds of expectations and variances: $\mathbb{E}[\mathbf{y}]$ and $Var[\mathbf{y}]$ for $\{\mathbf{y}, \mathbf{z}\}$, and $2\mathbb{E}[\mathbf{y}]$ and $4Var[\mathbf{y}]$ for $\{\mathbf{y} + \mathbf{z}\}$. Similarly, in the case where $m \in [1, n]$, there will be $m$ kinds of expectations and variances.

## C.3 Time Complexity Analysis

Following the convention [6, 52], we compare the time complexity of LMSPS with HAN [48], Simple-HGN [34], and SeHGNN [52] under mini-batch training with the total $N$ target nodes. All methods employ $l$-hop neighborhood. For simplicity, we assume that the number of features is fixed to $F$ for all layers. The average degree of each node is $rd$, where $r$ is the number of edge types and $d$ is the number of edges connected to the node for each edge type. The complexity analysis is

Table 9: Meta-paths searched by LMSPS on different datasets.

| Dataset | Meta-paths learnt by LMSPS |
|---|---|
| DBLP | AP, APT, APVP, APAPA, APTPA, APTPT, APTPV, APVPA, APVPV, APAPAP, APAPTP, APAPVP, APTPTP, APTPVP, APVPTP, APAPAPV, APAPTPA, APAPTPV, APAPVPA, APAPVPT, APAPVPV, APTPAPA, APTPAPT, APTPTPT, APTPVPV, APVPAPT, APVPTPA, APVPVPA, APVPVPT, APVPVPV |
| IMDB | M, MA, MK, MAM, MDM, MKM, MAMK, MDMK, MKMD, MDMKM, MKMAM, MKMDM, MKMKM, MAMAMK, MAMDMA, MAMDMK, MAMKMD, MAMKMK, MDMAMA, MDMAMD, MDMDMA, MDMDMK, MDMKMA, MKMAMA, MKMAMD, MKMAMK, MKMDMA, MKMDMK, MKMKMD, MKMKMK |
| ACM | PPP, PAPP, PCPA, PCPP, PPPC, PPPP, PAPAP, PAPPP, PCPAP, PCPPA, PPAPA, PPAPC, PPAPP, PAPAPA, PAPCPA, PAPPAP, PAPPCP, PAPPPP, PCPAPP, PCPCPP, PCPPAP, PCPPPP, PPAPAP, PPAPCP, PPAPPA, PPAPPP, PPCPAP, PPPAPA, PPPCPA, PPPPPP |
| OGBN-MAG | PF, PAPF, PFPP, PAPPP, PFPFP, PPAPP, PPPAP, PPPFP, PAIAPP, PAPAPF, PFPAPF, PFPPPF, PFPPPP, PPAPPF, PPPAPF, PPPPAP, PPPPPF, PAIAPAP, PAPAPAP, PAPAPPP, PAPPPAI, PAPPPPF, PAPPPPP, PFPAPAP, PFPPFPF, PFPPPPF, PPAPAPF, PPAPAPP, PPPAIAI, PPPAPPP |

summarized in Table 8. Because HAN and Simple-HGN require neighbor aggregation during training, and the number of neighbors grows exponentially with hops. So, they have neighbor aggregation costs of $O((rd)^l)$. SeHGNN employs a pre-processing step to avoid the training cost of neighbor aggregation. However, the exponential meta-paths cause SeHGNN to suffer from $O(r^l)$ costs in semantic aggregation. Unlike the above methods, LMSPS samples $M$ meta-paths in each iteration of the search stage and employs $M$ effective meta-paths in the training stage to avoid exponential costs. Generally, we have $O((rd)^l) \gg O(r^l) = O(K) \gg O(M)$ when maximum hop $l$ is large. The time complexity of LMSPS is a constant when $N$ and $F$ are determined, which is the key point for utilizing long-range meta-paths.

## D  Interpretability of Searched Meta-paths

We have conducted an extensive experimental study to validate the effectiveness of our searched meta-paths in the main text. Here, we illustrate the searched meta-paths of each dataset in Table 9. Because we discover many more meta-paths than traditional methods and most meta-paths are longer than traditional meta-paths, it is tough to interpret them one by one. So, we focus on the interpretability of meta-paths on large-scale `obgn-mag` dataset from the Open Graph Benchmark. The `OGBN-MAG` dataset is a heterogeneous graph composed of a subset of the Microsoft Academic Graph. It includes four different entity types: Papers (P), Authors (A), Institutions (I), and Fields of study (F), as well as four different directed relation types: Author $\xrightarrow{\text{writes}}$ Paper, Paper $\xrightarrow{\text{cites}}$ Paper, Author $\xrightarrow{\text{is affiliated with}}$ Institution, and Paper $\xrightarrow{\text{has a topic of}}$ Field. The target node is the paper, and the task is to predict each paper's venue (conference or journal).

Based on Table 9, the hop of effective meta-paths on `obgn-mag` ranges from 1 to 6, which means utilizing information from neighbors at different distances is important. Because long-range meta-paths provide larger receptive fields, LMSPS shows stronger capability in utilizing heterogeneity compared to traditional metapath-based HGNNs. The source node type of 16 meta-paths is P, *e.g.*, PFPFP (P←F←P←F←P). It indicates that the neighborhood papers of the target paper are most significant for predicting its venue, which is consistent with reality: the citation relationship, co-author relationship, and co-topic relationship between papers are usually the most effective information. 12 meta-paths' source node type is F. It implies that the neighborhood fields of the target paper are also crucial in determining its venue, which is also consistent with reality. Because most conferences or journals focus on a few fixed fields, a paper's venue is highly related to its field. The source node type of 2 meta-paths is I. It means the neighborhood institution is not very important for predicting the paper's venue, which is reasonable because almost all institutions have a wide range of conference or journal options for publishing papers. No meta-path has source node type A. It means the neighborhood author is unimportant in determining the paper's venue, which is logical because each paper has multiple authors, and each author can consider different venues. So, it is difficult to determine the paper's venue based on its neighborhood authors. If using too much institution or author information to predict the paper's venue, it actually introduces much useless information, which can be viewed as a kind of noise in `obgn-mag` for predicting each paper's venue.

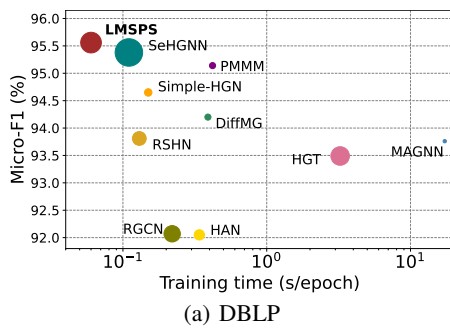
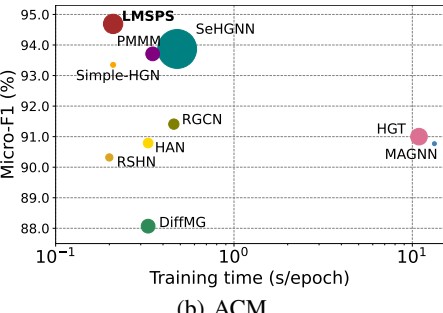



(a) DBLP                  (b) ACM



Figure 4: Micro-F1 scores, time consumption, and parameters of various HGNNs on `DBLP` and `ACM`. GTN has a large time consumption and parameters. We ignore it for ease of illustration.

Table 10: Performance of top mothod combinations on OGBN-MAG leaderboard.

| Method | OGBN-MAG | |
| --- | --- | --- |
| | Validation Accuracy | Test Accuracy |
| SAGN [42]+LP | $52.25 \pm 0.30$ | $51.17 \pm 0.32$ |
| GAMLP [58]+LP | $53.23 \pm 0.23$ | $51.63 \pm 0.22$ |
| SeHGNN+LP | $55.95 \pm 0.11$ | $53.99 \pm 0.18$ |
| LMSPS +LP | $\mathbf{56.98 \pm 0.10}$ | $\mathbf{55.10 \pm 0.11}$ |
| SeHGNN+LP+MS | $58.70 \pm 0.08$ | $56.71 \pm 0.14$ |
| SeHGNN+LP+MS+Emb | $59.17 \pm 0.09$ | $57.19 \pm 0.12$ |
| LMSPS +LP+MS | $\mathbf{59.51 \pm 0.07}$ | $\mathbf{57.84 \pm 0.22}$ |

We also provide some simple insights from the searced meta-paths of DBLP, IMDB, and ACM. In DBLP with target node type Author, the information from P (Paper) and A (Author) is slightly more important than that from T (Term) and V (Venue). In IMDB with target node type Movie, the importance of information of K (Keyword), M (Movie), A (Actor) and D (Director) gradually decreases. In ACM with target node type Paper, the importance of information of P (Paper), A (Author) and C (Conference) gradually decreases. For all datasets, the importance of node type is highly related to the target node type.

In addition, the hand-crafted meta-paths rely on intense expert knowledge, which is both laborious and data-dependent. In contrast, automatic meta-path search frees researchers from the understand-then-apply paradigm. In Table 9, LMSPS can search effective meta-paths without prior knowledge for various datasets, which is more valuable than manually-defined meta-paths.

# E More Experiments

## E.1 Comparison on Training Efficiency

Following the convention [34, 52], we show the time cost and parameters of LMSPS and the advanced baselines on `DBLP` and `ACM` in Figure 4. We measure the average time consumption of one epoch for each model. The area of the circles represents the parameters. The hidden size is set to $512$ and the maximum hop or layer is 6 for `DBLP` and 5 for `ACM` for all methods to test the training time and parameters under the same setting. Some methods perform quite poorly under the large maximum hop or layer. So we show the performance from Table 1 of the main text, which is the results under their best settings. Figure 4 shows that LMSPS has advantages in both training efficiency and performance. Our searched meta-paths are universal after searching once and can be applied to other metapath-based HGNNs based on Table 4. Because other metapath-based HGNNs don't include the time for discovering manual meta-paths, we also exclude our search time for discovering meta-paths in Figure 4. The search time is shown in Figure 3 (c) of the main text.

## E.2 Comparison with Top Method Combinations

The OGBN-MAG benchmark dataset is associated with a public leaderboard. In Table 10, we compare our method combination against top-performing approaches (method combinations) on the leaderboard. These methods integrate several general techniques, such as Label Propagation

Table 11: The performance of LMSPS under different searching epochs.

| Dataset | 1 | 5 | 10 | 20 | 40 | 60 | 80 | 100 | 120 | 140 | 160 | 180 | 200 |
|---|---|---|---|---|---|---|---|---|---|---|---|---|---|
| DBLP | 38.32 | 68.75 | 71.20 | 72.03 | 71.05 | 70.56 | 73.12 | 73.70 | 73.56 | 73.85 | 73.84 | 74.36 | 74.04 |
| IMDB | 43.48 | 49.78 | 53.88 | 58.48 | 57.03 | 58.40 | 58.05 | 58.65 | 57.54 | 58.70 | 58.59 | 58.72 | 58.81 |
| ACM | 79.03 | 85.43 | 87.42 | 86.75 | 88.30 | 89.85 | 88.96 | 89.40 | 88.52 | 89.85 | 89.70 | 89.85 | 89.91 |
| Freebase | 18.37 | 29.55 | 39.62 | 39.58 | 44.43 | 46.57 | 47.52 | 48.33 | 54.59 | 55.81 | 55.52 | 55.66 | 55.79 |
| OGBN-MAG | 14.21 | 26.20 | 32.41 | 37.00 | 40.82 | 45.89 | 41.74 | 44.00 | 48.40 | 48.55 | 48.70 | 48.43 | 48.59 |

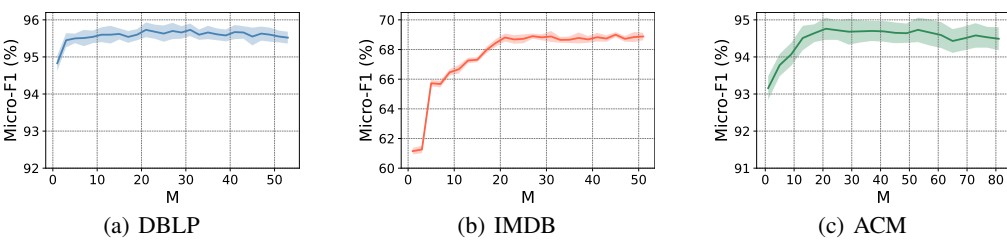

(a) DBLP      (b) IMDB      (c) ACM

Figure 5: Micro-F1 with respect to different hyper-parameter $M$ on DBLP, IMDB, and ACM.

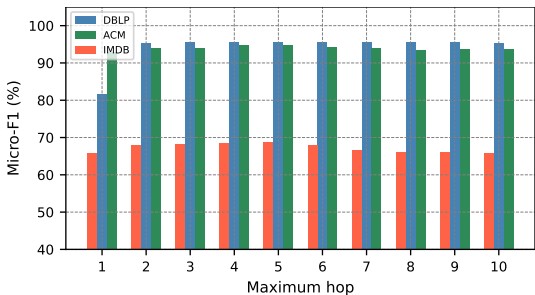

Figure 6: Micro-F1 scores with respect to different maximum hops on DBLP, IMDB, and ACM.

(LP), Multi-Stage Training (MS), and pre-trained Embeddings (Emb). By integrating LP and MS, LMSPS+LP+MS outperforms SeHGNN+LP+MS by a large margin. SeHGNN shows the results with extra embeddings as an enhancement. Though we cannot conduct a fair comparison with this trick due to the untouchability of its embedding file, LMSPS still outperforms their best result.

### E.3 Convergence of the Search Stage

As the search stage of LMSPS relies on sampling, we conduct experiments to explore the convergence of the search stage. Because the test set is unavailable in the search stage to avoid label leakage, we show the Micro-F1 scores of the validation set of DBLP, IMDB, ACM, Freebase and test accuracy of the validation set of OGBN-MAG under different searching epochs in Table 11. As we can see, LMSPS coverages during 100 to 180 epochs for all datasets. The main reason is that the search space size is progressively shrunk from $K$ to $2M$ under the progressive sampling algorithm. Most negative meta-paths have been removed from the search space in the second half of the search stage.

### E.4 Hyperparameter Study

LMSPS randomly samples $M$ meta-paths at each epoch in the search stage and selects the top-$M$ meta-paths in the training stage. Here, we perform analysis on hyper-parameter $M$ on DBLP, IMDB, and ACM. Freebase and OGBN-MAG are ignored because they are relatively large and the experiments are time-consuming. As illustrated in Figure 5, the performance of LMSPS increases with the growth of $M$ when $M$ is small. In addition, the performance on DBLP and ACM slightly decreases when $M$ is larger than a certain threshold, indicating excessive meta-paths don't benefit performance for some datasets. For unity, we set $M = 30$ for all datasets.

To observe the impact of different maximum hop values, we show the Micro-F1 of LMSPS with respect to different maximum hop values on DBLP, IMDB, and ACM in Figure 6. We can see LM-SPS show the best performance when the maximum hop is 5 or 6. Additionally, the performance of

LMSPS does not always increase with the value of the maximum hop, and the best maximum hop depends on the dataset.

## F    Reproducibility Statement

We have provided the details of datasets, tasks, baselines, and parameter settings in Appendix A and B and conducted the hyperparameter study in Appendix E.4. All reported results are the average of multiple experiments with standard deviations. We have included a pseudocode description of our method in Appendix C. The source code has been provided through an anonymized URL with clear commands on reproducing our results.

## G    Limitations

It is noteworthy that the performance of LMSPS does not always increase with the value of the maximum hop. For instance, based on Figure 3 (a), LMSPS can effectively utilize 12-hop meta-paths on DBLP with high performance and low cost. However, the optimal performance for LMSPS is observed when the maximum hop is 6. It can be attributed to the constraint of maintaining constant time complexity by setting the number of searched meta-paths to 30 for different maximum hop values. When the maximum hop value is 12, the number of target-node-related meta-paths exceeds 1400 (Figure 3 (a)). Searching for 30 effective meta-paths from such an extensive search space is notably challenging, despite LMSPS being the most effective method for meta-path search (as indicated in Table 3). The benefits of using longer meta-paths are outweighed by the drawbacks of the significantly more complex search space. So, the best maximum hop depends on the dataset and task and cannot be determined automatically.

Nevertheless, based on Table 8, the time complexity of LMSPS does not increase with the maximum hop. Consequently, it provides an effective solution for utilizing long-range dependency on heterogeneous graphs for possible applications on sparser real-world datasets. In Table 5, we conduct experiments on four constructed datasets based on OGBN-MAG to demonstrate that the advantages of utilizing long-range dependencies are more obvious for sparser large-scale heterogeneous graphs. For the constructed datasets, we carefully avoid inappropriate preference seed settings of randomly removing by limiting the maximum in-degree related to edge type. We expect sparser large-scale real heterogeneous datasets or more suitable tasks requiring long-range dependency to emerge in the future so we can fully explore our approach.

