# OpenReview forum: "Long-range Meta-path Search on Large-scale Heterogeneous Graphs"
_NeurIPS.cc/2024/Conference — NeurIPS 2024 poster_

### Official Review · Reviewer_r1gQ · 2024-07-08

**Soundness:** 2
**Presentation:** 3
**Contribution:** 2
**Rating:** 3
**Confidence:** 4

**Summary:**

This paper proposes an efficient meta-path search method on large-scale heterogeneous graphs. The proposed progressive sampling strategy and sampling evaluation strategy is effective for reducing the memory and time overhead, especially when the maximum hop is large. Experimental results show the effectiveness and efficiency of the proposed method.

**Strengths:**

1. The paper is well-written and easy to understand.
2. The experiments are comprehensive and the experimental results reveal the effectiveness of the proposed method.
3. The motivation sounds reasonable.

**Weaknesses:**

1. This novelty of this paper is limited.
2. The paper lacks a theoretical analysis about why the sampling strategy in the search stage is valid.
3. Experimental results on more large-scale graph datasets should be conducted.

**Questions:**

1. The differentiable search strategy is widely used in many automated graph learning tasks, such as DiffMG. There is no significant contribution in the search strategy other than the progressive sampling method. Moreover, the proposed sampling method is simply based on path strength. Similarly, the proposed sampling evaluation that uses multiple sampling to select top-$M$ meta-paths is also straightforward. Overall, the major concern about this paper is the limited novelty, although the experimental results seem good.

2. Why the simple progressive sampling method is effective? The authors should add a theoretical analysis about the effectiveness. For example, is it possible to filter out the promising meta-paths especially in the early stage of the search process?

3. The paper aims to achieve the meta-path search on large-scale datasets. However, only one large-scale dataset (i.e., OGBN-MAG) is employed. It is suggested to add more large-scale datasets to evaluate the effectiveness of the proposed method.

4. More experimental analysis should focus on large-scale datasets, such as the performance and efficiency analysis in Figure 3.

5. The interpretability meta-paths searched by LMSPS should be explained in detail. Many searched meta-paths seem strange and unreasonable. For example, PPPPPP in ACM, PPPAIAI in OGBN-MAG. The semantics of these meta-paths is hard to understand.

6. It is suggested to analyze all searched meta-paths in-depth and give more insights about how to choose suitable meta-paths manually.

7. Many hyperparameters need to be tuned on each dataset, such as the search space size $C$, the number of sampled meta-paths $M$, and the maximum hops.

---

> ### Author Rebuttal · Authors · 2024-08-06
>
> Thanks for your constructive comments. In the following, we respond to your concerns point by point.
>
> ---
>
> ### **W1: This paper has limited novelty because the proposed method is simple and straightforward.**
>
> **R1:** Although the other three reviewers enjoyed the novelty, we appreciate that the reviewer gave us the opportunity to highlight our novelty. We would like to clarify that the simplicity of the proposed LMSPS does not mean the novelty is limited. The novelty is highly related to the contribution. **If a method is effective,  a simple and straightforward design is better than a complex design**.
>
> The contributions of LMSPS have been summarized in the global response. To achieve the goals listed in the contributions, we propose a progressive sampling algorithm and a sampling evaluation strategy to overcome the efficiency and effectiveness challenges, respectively. **The high efficiency and high generalization of LMSPS exactly come from the simplicity** of the proposed method.
>
> In addition, compared to other differentiable search methods such as DiffMG, both the progressive sampling algorithm and sampling evaluation strategy are novel. In Table 3, LMSPS is compared with them and shows obvious advantages. We will clarify the novelty more clearly in the revised manuscript. Thank you very much!
>
> ---
>
> ### **W2: Why is the simple progressive sampling method effective?**
>
> **R2:** Thank you for the insightful question. As you mentioned, it is hard to filter out promising meta-paths in the early stage. So, as described in Lines 609-610 of Appendix B.4, LMSPS **warmups the parameters** for 20 epochs without dropping out any meta-paths.
>
> Because **the effectiveness of the algorithm is determined entirely by the final performance**, we can not provide theoretical analysis without experimental results to demonstrate the effectiveness. However, we have conducted adequate experiments to validate its effectiveness.
> * In Table 3, LMSPS is compared with six methods and shows obvious advantages.
> * In Table 6, the ablation study shows that both the progressive sampling method and sampling evaluation strategy can improve performance significantly.
> * In Table 11, the results show that the search stage of LMSPS can converge well on all five datasets.
>
> We believe these results have validated the effectiveness.  In addition, we have also provided an intuitive analysis of the effectiveness in Lines 634-639. Thanks to your question, we will highlight the effectiveness of the algorithm more clearly.
>
> ---
>
> ### **W3: It is suggested that more large-scale heterogeneous datasets be added.**
>
> **R3:** Thank you for the suggestion. Different from homogeneous graph fields, there are few academic large-scale datasets in heterogeneous graph fields due to heterogeneity. OGBN-MAG is the only large-scale heterogeneous graph dataset with a leaderboard and comparable baselines.
>
> To evaluate the effectiveness of LMSPS in more large-scale heterogeneous graphs, in Table 5, we also conduct experiments on four constructed datasets based on OGBN-MAG to demonstrate that the advantages of utilizing long-range dependencies are more obvious for sparser heterogeneous graphs. Based on your suggestion, if you could kindly provide some suitable large-scale heterogeneous datasets, we would love to evaluate LMSPS on them. Thank you!
>
> ---
>
> ### **W4:  More analysis, such as performance and efficiency analysis, should focus on large-scale datasets.**
>
> **R4:** Thank you for the suggestion. We would like to clarify that Table 2 and Lines 292-299 have analyzed the performance and efficiency of LMSPS on large-scale datasets under different maximum hops. In addition to Table 2, we also conducted a large number of experiments on large-scale datasets in Tables 1, 5, 6, 10, and 11. For most other experiments, we can only conduct reasonable comparisons on small and medium datasets **because most baselines run out of memory on large-scale heterogeneous graphs**. We will highlight the experiments and analysis of large-scale datasets in the revised manuscript. Thank you!
>
> ---
>
> ### **W5&W6: The interpretability of searched meta-paths should be explained. It is suggested to give more insights about how to choose meta-paths manually.**
>
> **R5&R6:** Thank you for the suggestion. One of the important features of meta-paths is their hop count. The 5-hop meta-path "PPPPPP" means aggregating the node features from 5-hop "P" neighbors through the paths "PPPPPP". We also provided a detailed example in the global response for understanding the searched meta-paths.
>
> The manual meta-paths rely on intense expert knowledge, which is both laborious and data/task-dependent. Considering it is impossible to understand the meta-paths of all real-world heterogeneous graphs, our automatic meta-path search shows great advantage by freeing researchers from the understand-then-apply paradigm.
>
> In lines 674-706, we have focused on explaining and giving some insights about choosing meta-paths in OGBN-MAG. We summarize them as: **The importance of information from P (Papers), F (Fields), I (Instructions)  to A (Authors)  gradually decreases**.
>
> The insights of the other datasets are shown in the global response. Due to space limitations, we can not interpret hundreds of meta-paths one by one here. We will add the explanations and insights in the revised manuscript. Thank you!
>
> ---
>
> ### **W7: Many hyperparameters need to be tuned on each dataset, such as C, M, and the maximum hops.**
>
> **R7:** Thank you for the question. We would like to clarify that **both C and M are not tuned on each dataset**. In Line 603, we have described that the number of selected meta-paths M is 30 for all datasets. Similarly, C is not tuned on each dataset. Based on Eq. (3), C is not a hyper-parameter but a dynamic value progressively decreasing with the search epochs. We will clarify them in the revised manuscript.
>
> Thank you once again for your insightful comments.

---

> > ### Comment · Reviewer_r1gQ · 2024-08-12
> > **Response to the rebuttal**
> >
> > Dear authors,
> >
> > Thanks for your rebuttal. My further concerns are as follows:
> >
> > 1. I recognize the statement that a simple and straightforward design is better than a complex design. But, I want to emphasize the novelty is rather incremental although the experimental results seem to be promising. Differentiable search method is widely used in many applications inspired by NAS (Neural Architecture Search). Applying differentiable search method to the meta-path search is not a novel idea. Moreover, some similar ideas for selecting meta-paths and reducing expert experiences have also been proposed. In other words, searching for meta-paths for heterogeneous graphs is not a new topic. Applying a commonly used search method for a not new topic is not very exciting. Although the authors states that the progressive sampling algorithm and sampling evaluation strategy are novel. But, from my view, the two tricks are too engineering without any theoretical analysis. And, the main body of this paper is still the differentiable search method. **Overall, the novelty of the paper is not yet up to level of NeurIPS**.
> > 2. The paper focus on the meta-path search on large-scale graphs. However, only one large-scale graph is used. Moreover, it is suggested that the authors can choose some large graphs from other fields not limited to academic graphs. **Many large graphs in OGB are heterogeneous and can be used for evaluation**.
> > 3. The concerns that many searched meta-paths seem strange and unreasonable still remains. Although the authors select some searched meta-paths and make explanations. But, most searched meta-paths lack interpretability. Interpreting the searched meta-paths one by one is infeasible. If the authors can find some common characteristics from the searched meta-path, this will have greater significance for guiding the design of the meta-path.
> >
> > Overall, due to the limited novelty, the lack of more large-scale graph datasets from different fields, and the lack of the interpretability and the insight about the searched meta-paths, I maintain my ratings unchanged.

---

> > > ### Author Response · Authors · 2024-08-13
> > > **Theoretical analysis about the reasonableness of sampling search**
> > >
> > > **Zero-order condition**: Consider two high-dimensional random variables $\mathbf{y} = f(\mathbf{x}) \in \mathbb{R}^{m \times d_1}$ and $\mathbf{z} = g(\mathbf{x}) \in \mathbb{R}^{m \times d_1}$. We say that $\mathbf{y}$ and $\mathbf{z}$ satisfy the zero-order condition if, for any valid sample $\mathbf{x} \in \mathbb{R}^{n \times d}$, the inequality $|\mathbf{y} - \mathbf{z}|_2 \leq \epsilon$ holds, where $\epsilon$ is a very small positive number.
> > >
> > > **Lemma 1**: Let $M$ represent the maximum number of activable paths, with each pair of operations satisfying the zero-order condition. Using $M$ distinct expectations and variances, it is possible to approximate all combinations (i.e., $2^M$).
> > >
> > > Lemma 1 guarantees that we can track the combination containing $i$ meta-paths with at least $M$ iterations. **Given that the number of iterations significantly exceeds $M$, the relative importance of the meta-paths can be learned during the search stage.**
> > >
> > > Below is the proof of Lemma 1:
> > >
> > > Let
> > >
> > > ${\mathbf{y}}_{p_y(y)} = f(\mathbf{x})$,
> > >
> > > ${\mathbf{z}}_{p_z(z)} = g(\mathbf{x})$,
> > >
> > > and ${\mathbf{x}} \sim p_{x(x)}$.
> > >
> > > For the case $M=1$, the expectations of $\mathbf{y}$ and $\mathbf{z}$ can be expressed as follows:
> > >
> > > $$
> > > \begin{aligned}
> > > {\\mathbb E}[{\\mathbf y}]&={\\mathbb E}[f\left({\\mathbf x}\right)]=\int p_{{\\mathbf x}}({\\mathbf x}) f({\\mathbf x})  {\mathrm d}{\\mathbf x}
> > > \end{aligned}
> > > $$
> > > $$
> > > \begin{aligned}
> > > {\\mathbb E}[{\\mathbf z}]&={\\mathbb E}[g\left({\\mathbf x}\right)]=\int p_{{\\mathbf x}}({\\mathbf x}) g({\\mathbf x})  {\mathrm d}{\\mathbf x}
> > > \end{aligned}
> > > $$
> > >
> > > According to the zero-order condition, $f(\mathbf{x}) \approx g(\mathbf{x})$. Since $p(\mathbf{x})$ is the same for both $\mathbf{y}$ and $\mathbf{z}$, it follows that $\mathbb{E}[\mathbf{y}] \approx \mathbb{E}[\mathbf{z}]$.
> > >
> > > Next, we prove that $Var[\mathbf{y}] \approx Var[\mathbf{z}]$. Note that $Var[\mathbf{y}] = \mathbb{E}\left[\mathbf{y}^{2}\right] - \left(\mathbb{E}[\mathbf{y}]\right)^{2}$ and $Var[\mathbf{z}] = \mathbb{E}\left[\mathbf{z}^{2}\right] - \left(\mathbb{E}[\mathbf{z}]\right)^{2}$. Thus, it suffices to prove that $\mathbb{E}\left[\mathbf{y}^{2}\right] \approx \mathbb{E}\left[\mathbf{z}^{2}\right]$. This can be similarly demonstrated as follows:
> > >
> > > $$
> > > \begin{aligned}
> > > {\\mathbb E}\left[{\\mathbf y}^{2}\right]&=\int p_{{\\mathbf y}}({\\mathbf y}) {\\mathbf y}^{2} {\mathrm d}  {\\mathbf y} = \int p_{{\\mathbf x}}({\\mathbf x}) f^{2}({\\mathbf x}) {\mathrm d} {\\mathbf x}
> > > \end{aligned}
> > > $$
> > >
> > > $$
> > > \begin{aligned}
> > > {\\mathbb E}\left[{\\mathbf z}^{2}\right]&=\int p_{{\\mathbf z}}({\\mathbf z}) {\\mathbf z}^{2} {\mathrm d}  {\\mathbf z} = \int p_{{\\mathbf x}}({\\mathbf x}) g^{2}({\\mathbf x}) {\mathrm d} {\\mathbf x}
> > > \end{aligned}
> > > $$
> > >
> > >
> > > According to the zero-order condition, we have $Var[\mathbf{y}] \approx Var[\mathbf{z}]$.
> > >
> > > For the case of $M=2$, when both paths are selected, the output becomes $\mathbf{y} + \mathbf{z}$, and its expectation can be written as:
> > >
> > > $$
> > > \begin{aligned}
> > > {\\mathbb E}[{\\mathbf y} + {\\mathbf z}] &= {\\mathbb E}[{\\mathbf y}] + {\\mathbb E}[{\\mathbf z}] \approx 2{\\mathbb E}[{\\mathbf y}]
> > > \end{aligned}
> > > $$
> > >
> > >
> > > The variance of $\mathbf{y} + \mathbf{z}$ is:
> > > $$
> > > \begin{aligned}
> > > Var[{\\mathbf y} + {\\mathbf z}] \approx  Var [2{\\mathbf y}] = 4 Var[\\mathbf y]
> > > \end{aligned}
> > > $$
> > >
> > > Thus, there are two types of expectations and variances: $\mathbb{E}[\mathbf{y}]$ and $Var[\mathbf{y}]$ for ${\mathbf{y}, \mathbf{z}}$, and $2\mathbb{E}[\mathbf{y}]$ and $4Var[\mathbf{y}]$ for ${\mathbf{y} + \mathbf{z}}$. Similarly, for the case where $M \in [1, K]$, there will be $M$ types of expectations and variances.

---

> ### Author Response · Authors · 2024-08-12
> **Thank you for the detailed response!**
>
> Thank you very much for the detailed response. In the following, we respond to your concerns point by point.
>
> ---
>
> ### **W1: Searching for meta-paths for heterogeneous graphs is not a new topic. Applying a commonly used search method for a not new topic is not very exciting.**
>
> **R1:** We appreciate that the reviewer gave us the opportunity to highlight our novelty again. We agree that searching for meta-paths for heterogeneous graphs is not a new topic. However, we would like to clarify that:
> 1. LMSPS is the first HGNN that makes it possible to achieve automated meta-path search for large-scale heterogeneous graph node property prediction*.
> 2. LMSPS is the first HGNN to utilize long-range dependency in large-scale heterogeneous graphs.
> 3. The searched meta-paths of LMSPS can be generalized to other HGNNs to boost their performance, which has not been achieved by existing works.
>
> **All the above contributions are new topics.** In addition, as described by the reviewer, the differentiable search method is widely used in many applications inspired by NAS. There have been dozens of papers accepted by top conferences. In our paper, we have cited the related paper a dozen times, highlighted the differences between LMSPS and them, and compared LMSPS with the representative methods. **Overall, we can conclude that our search method is totally new.** The theoretical analysis of the reasonableness of the sampling search is provided below. Thank you very much!
>
> ---
>
> ### **W2: Many large graphs in OGB are heterogeneous and can be used for evaluation**.
>
> **R2:** Thank you for the question. **We would like to clarify that many excellent works [1-4] also highlight their advantage on large-scale heterogeneous node property prediction only based on the results on OGBN-MAG.** In addition, in OGB, there are only six datasets for node property prediction: ogbn-products, ogbn-proteins, ogbn-arxiv, ogbn-papers100M, and ogbn-mag, and MAG240M. Except for ogbn-mag and MAG240M, all other four datasets are not heterogeneous. Because MAG240M contains over 240,000,000 nodes, it has hardly been tested by related works. The following tables shows the statistics of ogbn-mag and MAG240M. Although we have tried to run MAG240M, our hardware conditions cannot support the training even in the preprocessing stage. Thank you very much!
>
> | Dataset  | Num paper nodes | Num author nodes | Num institution nodes | Num field nodes | Total               |
> | -------- | --------------- | ---------------- | --------------------- | --------------- | ------------------- |
> | ogbn-mag | 736,389         | 1,134,649        | 8,740                 | 59,965          | 1,939,743           |
> | MAG240M  | 121,751,666     | 122,383,112      | 25,721                | -               | 244,160,652  (126x) |
>
> | Dataset  | Num paper-paper edges | Num author-paper edges | Num author-institution edges | Num paper-field edges | Total                |
> | -------- | --------------------- | ---------------------- | ---------------------------- | --------------------- | -------------------- |
> | ogbn-mag | 5,416,271             | 7,145,660              | 1,043,998                    | 7,505,078             | 21,111,007           |
> | MAG240M  | 1,297,748,926         | 386,022,720            | 44,592,586                   | -                     | 1,728,364,232  (82x) |
>
> ---
>
> ### **W3: The authors should find some common insights from the searched meta-path.**
>
> **R3:** We appreciate that the reviewer gave us the opportunity to highlight our insights again. We have shown insights into DBLP, IMDB, and ACM in the global response. We show them again as follows.
>
> * In DBLP with target node type Author, the information from P (Paper) and A (Author) is slightly more important than that from T (Term) and V (Venue).
> * In IMDB with target node type Movie, the importance of information of K (Keyword), M (Movie), A (Actor) and D (Director) gradually decreases.
> * In ACM with target node type Paper, the importance of information of P (Paper), A (Author) and C (Conference) gradually decreases.
> * **For all related datasets, the importance of node type is highly related to the target node type**.
>
> ---
>
> Thank you once more for your efforts. Please kindly let us know if our response has addressed your concerns. We are happy to answer your remaining concerns and questions if you have any.
>
> ---
>
> [1] Open Graph Benchmark: Datasets for Machine Learning on Graphs. NeurIPS, 2020.
>
> [2] Graph Attention Multi-Layer Perceptron. KDD, 2022.
>
> [3] Simple and Efficient Heterogeneous Graph Neural Network. AAAI, 2023.
>
> [4] An Efficient Subgraph-Inferring Framework for Large-Scale Heterogeneous Graphs. AAAI, 2024.

---

### Official Review · Reviewer_71Wo · 2024-07-09

**Soundness:** 3
**Presentation:** 4
**Contribution:** 3
**Rating:** 7
**Confidence:** 4

**Summary:**

The paper proposes a novel framework LMSPS, aimed at efficiently utilizing long-range dependencies in large-scale HIN. The framework addresses two primary challenges: reducing computational costs while maximizing information utilization and overcoming the over-smoothing problem common in GNNs. LMSPS employs a progressive sampling algorithm to dynamically reduce the search space for meta-paths, thus identifying a subset of effective meta-paths tailored to the specific dataset and task.

**Strengths:**

1. The paper is well-written and easy to follow. The technical designs are clearly described.
2. The idea of using a progressive sampling algorithm to narrow the search space for meta-paths is novel and well motivated.
3. The framework is designed to handle large-scale graphs efficiently, maintaining stable performance and resource usage even as the complexity of meta-paths increases.
4. The proposed method consistently outperforms SOTA baselines across multiple datasets, including large-scale datasets like OGBN-MAG, demonstrating its robustness and effectiveness.

**Weaknesses:**

1. I still have some doubts about the necessity of modeling long-range dependencies in heterogeneous graphs. For example, in an academic network, the label of a paper can be well predicted by relying on some very close nodes. Could the authors provide an example to illustrate specific situations where long-range dependency is crucial?
2. Although the paper compares LMSPS with various baselines, more detailed ablation studies focusing on the individual components of LMSPS would strengthen the validation of its effectiveness.

**Questions:**

See W1.

**Limitations:**

NA.

---

> ### Author Rebuttal · Authors · 2024-08-06
>
> Thanks for your positive comments that greatly encourage us. In the following, we respond to your concerns point by point.
>
> ---
>
> ### **W1: I still have some doubts about the necessity of modeling long-range dependencies in heterogeneous graphs. Could the authors provide an example to illustrate specific situations where long-range dependency is crucial?**
>
> **R1:**  Thank you so much for the insightful question. As shown in Table 2 of Section 6.3, SeHGNN can not utilize meth-paths larger than three hops on OGBN-MAG, and the best performance is 52.44%, while the performance of LMSPS increases from 52.72% to 54.83% when the maximum hop grows from 3 to 6. We can see that long meta-paths can increase performance significantly, but the performance of short meta-paths is also not bad. Utilizing long meta-paths means freely combining effective information from long and short meta-paths, which is the core advantage of LMSPS.
>
> Understanding long meta-paths is difficult. However, automatic meta-path search actually frees researchers from the understand-then-apply paradigm. Based on LMSPS, we can search long and effective meta-paths without prior knowledge for various datasets. It is much more convenient than defining manual meta-paths based on expert knowledge, which is both laborious and data/task-dependent.
>
> Thanks to your suggestion, we take the meta-path MDMDMK (M←D←M←D←M←K)  from IMDB as an example. IMDB includes four different entity types: Movies (M), Directors (D), Keywords (K), and Actors (A). The task is to predict the category of the target movies.   MDMDMK is a 5-hop meta-path that is hard for experts to understand and then apply. However, for many movies without keywords, the meta-path M←D←D←D←M←K is important because the target movies can aggregate the keyword information from the movies of co-directors. This example shows **the ability of long-range dependencies to complete the missing information that can not be obtained from close nodes**.
>
> We hope the example meets your requirements. We will clarify the necessity of modeling long-range dependencies more clearly in the revised manuscript. Thank you!
>
> ---
>
> ### **W2: More detailed ablation studies focusing on the individual components of LMSPS would strengthen the validation of its effectiveness.**
>
> **R2:** Thank you for the kind suggestion. In Table 6 of Section 6.6, we have conducted ablation studies to analyze the effects of individual components in LMSPS. Specifically, we separately remove the progressive sampling method, sampling evaluation strategy, and concatenation operation to observe the performance of LMSPS. As shown in Table 6, the performance of LMSPS significantly decreases when removing progressive sampling or sampling evaluation strategy and slightly decreases after replacing the concatenation operation with the transformer block. We will highlight the ablation study in the revised manuscript. Thank you!

---

### Official Review · Reviewer_msas · 2024-07-12

**Soundness:** 3
**Presentation:** 4
**Contribution:** 3
**Rating:** 7
**Confidence:** 4

**Summary:**

The paper proposes a new framework called Long-range Meta-path Search through Progressive Sampling (LMSPS), which differs from traditional meta-path-based GNN training methods on heterogeneous graphs. LMSPS introduces a strategy for building a search space that includes all meta-paths related to the target node type and employs a sampling evaluation strategy to conduct specialized and effective meta-path selection.
The authors studied two new observations: a) A small number of meta-paths dominate the performance, b) Certain meta-paths can have a negative impact on performance.
They designed the LMSPS framework with a super-net in the search stage and a target-net in the training stage to mitigate costs and the over-smoothing problem that occurs in recent Heterogeneous Graph Neural Networks.
In conclusion, the paper presents a new framework that could help us better understand the challenges of leveraging long-range dependencies in large-scale heterogeneous graphs.

**Strengths:**

* The paper in general presents a nice framework. The paper is well-written, has good coherence, and is well-structured.
* The two observations are interesting, i.e., a few meta-paths dominate the performance, and certain meta-paths can have negative impact on performance.
* The paper is very clear with thorough experiments and analysis. The originality of the experiment is strong.
* The experiments have demonstrated their limitations and possibilities to improve the effectiveness across datasets.

**Weaknesses:**

* Even though the observations are interesting, I think the paper could do more to explore their implications for robust generalization. What does robust generalization to other classes imply or reveal about the process of long-range meta-path GNN training? The reduction of meta-path samples is effective. The authors clarify that the optimal maximum hop depends on the dataset and task, as mentioned in Appendix G, and cannot be determined automatically. Thus, will the sampling search impact the model's robustness?
* The evaluation results observed in Figure 1(a) and (b) cannot demonstrate the improvement in performance upon the removal of certain meta-paths. This raises the question of whether the observation that "certain meta-paths can have a negative impact on performance" is generally applicable. The second observation from Figure(c) is derived from the ACM dataset. Will this observation be valid for other heterogeneous graphs?
* What is the cost of the pre-processing stage? Can it be reduced by sampling to decrease the neighbor aggregation costs?
* Line 172 clarifies that MLP requires less human intervention compared to Transformer. Are there specific experiments demonstrating the superiority of MLP?

**Questions:**

Please see the weakness section.

**Limitations:**

The authors have adequately addressed the limitations of their work.

---

> ### Author Rebuttal · Authors · 2024-08-06
>
> Thanks for your positive comments that greatly encourage us. In the following, we respond to your concerns point by point.
>
> ---
>
> ### **W1:  What does robust generalization to other classes imply or reveal about the process of long-range meta-path GNN training? Will the sampling search impact the model's robustness on searched meta-paths?**
>
> **R1:**   Thank you for the insightful question. As described in Lines 170-171,  to discover meta-paths with high generalization,  the search results should not be affected by specific modules. So, the more complex the model, the harder it is for the discovered meta-paths to generalize well, which can be viewed as a kind of overfitting on architectures. Based on this consideration, compared to the previous works [1-3] that do not show the generalization, the architecture of our model is very simple, with pure MLPs as the parametric modules.
>
> The sampling strategy is very important for our meta-paths search. Its function can be summarized as follows.
>
> * Similar to dropout, the sampling strategy keeps the parametric modules changing in the search stage, which is important for preventing the search meta-paths from being affected by specific modules. **So, the sampling search can increase the generalization ability.**
> * When the maximum hop is large, the search stage will run out of memory without the sampling strategy.
> * The sampling strategy can overcome the deep coupling issue [4] of differentiable search by introducing randomness in each iteration.
>
> As the search stage aims to determine the relative importance of meta-paths rather than achieve robust accuracy, the sampling search has little impact on the robustness of the searched meta-paths. Specifically, because the search stage has many iterations, the architecture parameter of each meta-path will be updated multiple times and the relative importance can be learned during training even with the sampling strategy. It is supported by Table 11 in Appendix E.3, which shows that the search stage of LMSPS can converge well on all five datasets. Thanks to the question, we will clarify the corresponding part more clearly in the revised manuscript.
>
> ---
>
> ### **W2:  Will the second observation, "certain meta-paths can have a negative impact for heterogeneous graphs", be valid for other heterogeneous graphs?**
>
> **R2:**  Thank you for the insightful question. Except for ACM, in Lines 142-143, we have described an important fact that various recent HGNNs [5-7] have removed some edge types in the Freebase dataset to exclude corresponding heterogeneous information during pre-processing based on substantial domain expertise or empirical observations. The basic logic behind this behavior is that the meta-paths related to the edge types have negative impacts on heterogeneous graphs in the task. In addition, in Figure 5 of Appendix E.4, we can see that the performance of LMSPS doesn't always increase with the growth of the number of utilized meta-paths on DBLP, IMDB, and ACM, which also supports the second observation to some extent. Thanks to the question, we will clarify the corresponding part more clearly.
>
> ---
>
> ### **W3: What is the cost of the pre-processing stage? Can it be reduced by sampling to decrease the neighbor aggregation costs?**
>
> **R3:** Thank you for the questions. Following SeHGNN [7], the pre-processing executes the simplified neighbor aggregation only once without any parameter updating. Specifically, we use the multiplication of adjacency matrices to calculate the final contribution weight of each node to targets without calculating nodes in the middle of paths, which is much more efficient than the pre-processing step in HAN [3]. On small datasets, the pre-processing stage takes 1~4 seconds. On OGBN-MAG, the pre-processing stage takes about 130 seconds, which is about 1.2% of the whole training stage. We believe the neighbor aggregation costs can be reduced by sampling. However, it is not very necessary because the current pre-processing cost is much smaller than the training cost. We will explore this in future work and clarify the corresponding part more clearly in the revised manuscript. Thank you!
>
> ---
>
> ### **W4: Line 172 clarifies that MLP requires less human intervention compared to Transformer. Are there specific experiments demonstrating the superiority of MLP to Transformer?**
>
> **R4:** Thank you for the question. Although the Transformer is a widely used and powerful model, in line 172, we highlight that the Transformer involves more inductive bias, i.e., human intervention, than MLPs, which have been supported by many works [8-10].
>
> In LMSPS, the difference of importance between the searched effective meta-paths is much smaller than that between the full meta-path set, making the attention mechanism seem unnecessary. For higher efficiency and generalization, we use pure MLPs instead of Transformer. In Table 6 of the ablation study, we employ the Transformer for semantic attention on all meta paths. The Transformer version performs slightly worse than LMSPS even if it uses many more meta-paths and is out-of-memory on Freebase and OGBN-MAG. We will clarify the corresponding part more clearly in the revised manuscript. Thank you very much!
>
> ---
>
> **References:**
>
> [1] Graph Transformer Networks. NeurIPS, 2019.
>
> [2] Heterogeneous Graph Transformer. WWW, 2020.
>
> [3] Heterogeneous Graph Attention Network. WWW, 2022.
>
> [4] Single Path One-Shot Neural Architecture Search with Uniform Sampling. ECCV, 2020.
>
> [5] DiffMG: Differentiable Meta Graph Search for Heterogeneous Graph Neural Networks. KDD, 2021.
>
> [6] Differentiable Meta Multigraph Search with Partial Message Propagation on Heterogeneous Information Networks. AAAI, 2023.
>
> [7] Simple and Efficient Heterogeneous Graph Neural Network. AAAI, 2023.
>
> [8] MLP-Mixer: An all-MLP Architecture for Vision. NeurIPS, 2021.
>
> [9] A Generalization of ViT/MLP-Mixer to Graphs. ICML, 2023.
>
> [10] Scaling MLPs: A Tale of Inductive Bias. NeurIPS, 2024.

---

> > ### Comment · Reviewer_msas · 2024-08-12
> >
> > Thank you for your response. These have addressed the majority of my questions. I appreciate the effort and insights the authors put into the paper. I will maintain my original score.

---

> > > ### Author Response · Authors · 2024-08-12
> > >
> > > Thank you for appreciating the effort and insights! We're glad to hear that you're satisfied.

---

### Official Review · Reviewer_dJZo · 2024-07-14

**Soundness:** 2
**Presentation:** 3
**Contribution:** 2
**Rating:** 5
**Confidence:** 4

**Summary:**

This paper presents an empirical study demonstrating that not all meta-paths are useful; some even negatively impact performance. Selecting the most meaningful meta-paths is crucial. The authors propose LMSPS, a super-net-based method to select beneficial meta-paths effectively.

**Strengths:**

S1. The presentation is excellent and easy to follow.
S2. This paper is the first attempt to combine super-net and heterogeneous graph learning.
S3. Experimental results show that their model achieves state-of-the-art (SOTA) performance.

**Weaknesses:**

W1. The paper's title is misleading. According to the title, the work seems to search for meaningful long-range meta-paths only to improve the performance of HIN representation learning. However, I think the major idea is to select effective meta-paths efficiently to overcome the issue of the exponential increase in the number of meta-paths. As shown in Table 9, some short meta-paths are still important. This work is actually a meta-path selection task in my point of view.

W2. The motivation of SeHGNN is that "models with a single-layer structure and long meta-paths outperform those with multi-layers and short meta-paths". I think it does not mean that long-range meta-paths are more important than short paths. I agree that different meta-paths have different importance. But is the length of the paths the main reason for this? As analyzed in the Limitation section, although the maximum hop is set to 12, the best performance is achieved at 6. Some early studies claimed that long paths can introduce noise and less relevant connections between nodes, leading to less accurate or meaningful representations.

W3. For most datasets, the improvement is marginal compared to the second-best performance. The enhancement brought by the selected meta-paths is not significant enough according to the experimental results.

**Questions:**

1. Please refer to the aforementioned weaknesses.
2. In OGB, I can see that LMSPS achieved the 3rd place. Have you tried to compare with the first two methods?

**Limitations:**

Limitations are sufficiently stated in the appendix. To my knowledge, there is no negative societal impact of this work.

---

> ### Author Rebuttal · Authors · 2024-08-06
>
> Thanks for your constructive comments. In the following, we respond to your concerns point by point.
>
> ---
>
> ### **W1: The paper's title is misleading. I think the major idea is to select effective meta-paths efficiently to overcome the issue of the exponential increase in meta-paths.**
>
> **R1:**  Thank you for the careful reading and insightful comments. On homogeneous graph fields, there have been many outstanding works [1-4] that highlight long-range dependencies in their title but still employ short-range dependencies. We think that the logic behind the behavior is **the importance of short-range dependencies is widely accepted and utilizing them is not challenging**. So, they highlight long-range dependencies in their title. Similarly, although our work can search meaningful short meta-paths, the most significant contribution differentiating this work from others is the ability to search effective long-range meta-paths in heterogeneous graphs.
>
> Moreover, though the major idea is totally the same as you mentioned, the issue of exponential increase in the number of meta-paths comes from our attempt to search long-range meta-paths. So, utilizing long-range dependency in heterogeneous graphs is the purpose, and overcoming the exponential issue is the specific process.
>
> If the reviewer could kindly provide a more suitable title, we would love to use it. Thank you very much!
>
> ---
>
> ### **W2: I think SeHGNN does not mean that long-range meta-paths are more important than short paths. Long paths can introduce noise and less relevant connections between nodes, leading to less accuracy.**
>
> **R2:** Thank you. This is a very insightful comment. Although we also do not think long-range meta-paths are more important than short meta-paths, we appreciate that the reviewer gave us the opportunity to explain our key idea of utilizing long meta-paths. Compared to existing work only utilizing short meta-paths, the key advantage of LMSPS is freely combining the effective information from long and short meta-paths
>
> **Although most long meta-paths can introduce noise or redundant information, some effective long meta-paths can bring extra valuable information beyond short meta-paths**. The exact example can be found in the global response. Searching for effective long-range meta-paths is exactly one of our key contributions. As shown in Table 2 of Section 6.3, the performance of SeHGNN [5] and LMSPS keeps increasing as longer meta-paths are gradually introduced. However, SeHGNN can not utilize meth-paths larger than three hops on OGBN-MAG and the best performance is 52.44%, while the performance of LMSPS increases from 52.72% to 54.83% when the maximum hop grows from 3 to 6. So, utilizing effective long meta-paths can improve performance instead of leading to less accuracy. We will clarify it more clearly in the revised manuscript. Thank you very much!
>
> ---
>
> ### **W3: The enhancement is not significant enough according to the experimental results.**
>
> **R3:** Although the other three reviewers highlighted the enhancement of experimental results in Strengths, we appreciate that the reviewer allowed us to explain our results more clearly. Based on Table 1, most of the second-best results come from SlotGAT [6]. LMSPS achieves an average of 1.00% absolute improvement compared to SlotGAT on small and medium datasets. Considering these datasets are widely used and the existing scores are high enough, a 1.00% average improvement is not bad. Moreover, SlotGAT can not run on OGBN-MAG due to the out-of-memory issue, highlighting the advantage of LMSPS.
>
> The second-best result on OGBN-MAG is 51.45%, which is outperformed by LMSPS by a large margin of 3.38%. Considering that OGBN-MAG is a large-scale dataset that is much more challenging than the other datasets and LMSPS is designed for large-scale heterogeneous graphs, the improvements can validate its effectiveness. In addition, based on Table 5, LMSPS outperforms the second-best method by a large margin of 4.78% on the sparser large-scale dataset, which is also significant. We will explain our results more clearly in the revised manuscript. Thank you very much!
>
> ---
>
> ### **W4: LMSPS achieves the 3rd place in ogbn-mag leaderboard. Have you tried to compare with the first two methods?**
>
> **R4:** Thank you for the question. I have tried to compare LMSPS with the first two methods, which use curriculum learning to change the input sequence of the data. However, we notice their results have been widely questioned due to the test label leakage problem (The code for the first place is completely based on that for the second place). The OGB team also noticed this problem and has asked the authors to investigate the results in two weeks. However, the authors haven't finished the investigation yet and have asked the OGB team to remove the leaderboard submission. Although we can not provide the external link due to the rebuttal policy, the discussion can be easily searched on GitHub. In addition, neither of the above works has complete papers, making their methods less trustworthy.
>
> In summary, **LMSPS still ranks 1st on the ogbn-mag with trustworthy results**.
>
> Thank you once again for your insightful comments.
>
> ---
>
> **References:**
>
> [1] Representing Long-Range Context for Graph Neural Networks with Global Attention. NeurIPS, 2021.
>
> [2] Graph-based high-order relation modeling for long-term action recognition. CVPR, 2021.
>
> [3] Hope: High-order graph ode for modeling interacting dynamics. ICML, 2023.
>
> [4] High-order pooling for graph neural networks with tensor decomposition. NeurIPS, 2022.
>
> [5] Simple and Efficient Heterogeneous Graph Neural Network. AAAI, 2023.
>
> [6] SlotGAT: Slot-based Message Passing for Heterogeneous Graphs. ICML, 2023.

---

> ### Author Response · Authors · 2024-08-13
> **Looking forward to your response**
>
> Dear Reviewer #dJZo,
>
> We sincerely appreciate your thorough review and insightful comments on our manuscript. We have taken the time to carefully address all the points you raised, including
>
> - explaining the misleading title
> - clarifying the importance of long-range meta-paths and their role
> - defending the improvement, especially in the ogbn-mag leaderboard
> - explaining the leaderboard issues
>
> Please refer to the **rebuttal content** for details.
>
> As the discussion period is limited to 7 days and only 1 day remains, we kindly request your prompt feedback on our responses. Your expertise is crucial to us, and we welcome any additional thoughts you may have. Thank you once again for your time and attention.
>
> Best regards,
>
> Authors of #4444

---

> > ### Comment · Reviewer_dJZo · 2024-08-14
> >
> > Thank you for the detailed response. I am sorry for providing my feedback late.
> >
> > W3 and W4 have been addressed satisfactorily. Below are more comments regarding W1 and W2.
> >
> > W1: My further concern is that there is no analysis of how the discovered long-range meta-paths benefit performance. It would be interesting to see some discussions about the impact of the discovered long-range meta-paths on performance improvement. Additionally, the methodology seems to lack specific techniques for searching meaningful "long-range" meta-paths. It mainly focuses on utilizing a super-net to discover "useful" meta-paths. The novelty of this part seems marginal.
> >
> > W2: I think SeHGNN is primarily designed for efficient heterogeneous graph learning rather than discovering useful meta-paths. It cannot handle long-hop meta-paths due to the setting of enumerating all possible meta-paths. If this setting is changed to a limited number of meta-paths, SeHGNN could be efficient due to its simplified attention mechanism. Again, since the motivation of this work is to efficiently select the most effective meta-paths, the experiments should give more discussions about the quality of the discovered meta-paths.

---

> ### Author Response · Authors · 2024-08-14
> **Thank you for the detailed response!**
>
> Thank you very much for the detailed response. In the following, we respond to your concerns point by point.
>
> ### W1: How do the discovered long-range meta-paths benefit performance? The methodology seems to lack specific techniques for searching meaningful "long-range" meta-paths.
>
> **R1:** Thank you for the insightful question. In the global rebuttal "For the importance of long-range meta-paths," we have provided a detailed example to show the ability of long-range dependencies to complete the missing information that can not be obtained from close nodes. Also, as shown in Table 2 in Section 6.3 (also shown below for quick check),  the performance of LMSPS keeps increasing as the maximum hop value grows, which means gradually adding longer meta-paths. It indicates that LMSPS can overcome the issues caused by utilizing long-range dependency, e.g., over-smoothing and noise. Moreover, as shown in Table 4 of Section 6.4, the discovered meta-path can also benefit SeHGNN.
>
> Searching for long-range meta-paths has two main challenges: the exponentially increasing issue and the noise issue. To overcome both issues, as described in Lines 157-169, we propose a progressive sampling algorithm and a sampling evaluation strategy to overcome the two challenges, respectively. Specifically, the high-efficiency progressive sampling algorithm ensures LMSPS can search effective short and long-range meta-paths under a large maximum hop. As different meta-paths could be noisy or redundant to each other, top-M meta-paths are not necessarily the optimal solution when their importance is calculated independently. The sampling evaluation strategy evaluates the overall performance of each meta-path set. So, it can overcome the noise issue.
>
> | Max hop | Num path |  SeHGNN (Time / Test Acc (%))  |    LMSPS (Time / Test Acc(%))     |
> | :-----: | :------: | :------------------------: | :----------------------------: |
> |    1    |    4     | 4.35   /   47.18  |   3.98   /   46.88    |
> |    2    |    10    | 6.44   /   51.79  |   5.63   /   51.91    |
> |    3    |    23    | 11.28   /   52.44 |   10.02   /   52.72   |
> |    4    |    50    |            OOM             |   14.34   /   53.43   |
> |    5    |   107    |            OOM             |   14.77   /   53.90   |
> |    6    |   226    |            OOM             | 14.71   /   **54.83** |
>
> ---
>
> ### **W2.1: If this setting is changed to a limited number of meta-paths, SeHGNN could be efficient due to its simplified attention mechanism.**
>
> **R2.1:** Thank you for the insightful comment. Based on our second observation, i.e., certain meta-paths can have a negative impact on heterogeneous graphs; the attention mechanism has limitations in dealing with negative meta-paths. As described in Lines 142-144, the second observation is supported by the fact that various recent HGNNs have removed some edge types to exclude corresponding heterogeneous information during the pre-processing stage. For example, SeHGNN removes all edge types related to node type F (Field) in the ACM dataset. If simplified attention can handle negative meta-paths, this step is unnecessary. On the contrary, by meta-path search, LMSPS can easily drop out negative meta-paths. Table 2 supports this conclusion. When the maximum hop is 3, LMSPS outperforms SeHGNN by 0.28%, even the latter using more meta-paths.
>
>
>
> ---
>
>
>
> ### **W2.2: The experiments should give more discussions about the quality of the discovered meta-paths.**
>
> **R2.2:**  As described in Lines 310-319, to demonstrate the high quality of searched meta-paths, on the one hand, the meta-paths should be effective in the proposed model. On the other hand, the effective meta-paths mainly depend on the dataset instead of the architecture, so the meta-paths should be effective after being generalized to other HGNNs. Based on the results in Tables 1,2,3,5, using the search meta-paths, LMSPS outperforms the other baselines on almost all conditions, sometimes significantly, which can validate the high quality of the discovered meta-paths on the proposed model.
>
> Because finding meta-paths that work effectively across various HGNNs is a tough task, it has not been achieved by previous works. However, based on Table 4 (also shown below for quick check), After simply replacing the original meta-path set with our searched meta-paths and keeping other settings unchanged, the performance of HAN and SeHGNN both improve, demonstrating the effectiveness of our searched meta-paths. Thank you very much!
>
> | Method       | DBLP           | IMDB           | ACM            | Freebase       |
> | ------------ | -------------- | -------------- | -------------- | -------------- |
> | HAN          | 92.05 | 64.63 | 90.79 | 54.77 |
> | HAN-LMSPS    | 93.54 | 65.89 | 92.28 | 57.13 |
> | SeHGNN       | 95.24 | 68.21 | 93.87 | 63.41 |
> | SeHGNN-LMSPS | 95.57 | 68.59 | 94.46 | 65.37 |
>
> ---
>
> Thank you once more for your efforts. Please kindly let us know if our response has addressed your concerns.

---

### Author Rebuttal · Authors · 2024-08-06

We are very grateful to the reviewers for carefully reviewing our paper and providing constructive comments and suggestions that have helped improve our submission. We especially thank the reviews for recognizing that our paper has:

1. **good originality** on method ((Reviewers djZo and 71Wo) and experiments (Reviewer maas),
2. **outstanding experiment results** (All Reviewers),
3. **nice presentation** (All Reviewers).

The main concerns include the limited novelty (r1gQ), the third-place ranking in OGB (djZo), the importance of long-range meta-paths (dJZo, 71Wo), and the insights from searched meta-paths (r1gQ). We briefly introduce the responses to these concerns in this general response section and provide concrete details in the response to each reviewer.

**For the limited novelty**, because the novelty is highly related to the contributions. We summarize the contributions as follows.

- Large-scale dataset. LMSPS is the first HGNN that makes it possible to achieve automated meta-path selection for large-scale heterogeneous graph node property prediction.
- Long-range dependency. LMSPS is the first HGNN to utilize long-range dependency in large-scale heterogeneous graphs. To achieve the above two goals, LMSPS has addressed two key challenges: (1) Alleviating costs while striving to effectively utilize information in exponentially increased receptive fields and (2) overcoming the well-known over-smoothing issue.
- High generalization. As shown in Table 6, the searched meta-paths of LMSPS can be generalized to other HGNNs to boost their performance, which has not been achieved by existing works. To accomplish this objective, LMSPS uses an MLP-based architecture instead of a transformer-based architecture for meta-path search because the former involves fewer inductive biases, i.e., human interventions.

**For the third place ranking in OGB**, we have provided the details that the first two methods were questioned by the OGB team due to the test label leakage problem. The authors have asked the OGB team to remove the leaderboard submission.

**For the importance of long-range meta-paths**, we have provided a detailed example to show the ability of long-range dependencies to complete the missing information that can not be obtained from close nodes. Take the meta-path MDMDMK (M←D←M←D←M←K)  from IMDB as an example. IMDB includes four different entity types: Movies (M), Directors (D), Keywords (K), and Actors (A). The task is to predict the category of the target movies. MDMDMK is a 5-hop meta-path that is hard for experts to understand and then apply. However, for many movies without keywords, the meta-path M←D←D←D←M←K is important because the target movies can aggregate the keyword information from the movies of co-directors.

**For the insights from searched meta-paths**, we have added the missing insights from the searced meta-paths of DBLP, IMDB, and ACM. In DBLP with target node type Author, the information from P (Paper) and A (Author) is slightly more important than that from T (Term) and V (Venue). In IMDB with target node type Movie, the importance of information of K (Keyword), M (Movie),  A (Actor) and D (Director) gradually decreases. In ACM with target node type Paper, the importance of information of P (Paper), A (Author) and C (Conference) gradually decreases. In addition, the importance of node type is highly related to the target node type.

**We hope that our response has addressed your concerns. In case you still have some concerns or we missed anything, please let us know.**

**Best regards**

---

### Decision · Program_Chairs · 2024-09-25

**Decision:**

Accept (poster)

**Comment:**

This paper shows that a small number of meta paths are important for the performance, whereas certain meta-paths can negatively impact it. In light of this, the paper proposes an approach that makes meta path selection via a sampling evaluation strategy. Most reviewers recognize that this paper is well-written, the experiments are strong, and the results are useful. The authors should carefully address all the concerns of the reviewers.